# ProDOL: a general method to determine the degree of labeling for staining optimization and molecular counting

Stanimir Asenov Tashev[1,2,3,10], Jonas Euchner [1,2,3,4,10], Klaus Yserentant [1,2,3,4,8,10], Siegfried Hänselmann [4], Felix Hild[4], Wioleta Chmielewicz[4,5], Johan Hummert[1,2,3,4,9], Florian Schwörer [6], Nikolaos Tsopoulidis [5], Stefan Germer[4], Zoe Saßmannshausen [1,7], Oliver T. Fackler [5], Ursula Klingmüller[6] & Dirk-Peter Herten [1,2,3,4] ✉

Determining the label to target ratio, also known as the degree of labeling (DOL), is crucial for quantitative fluorescence microscopy and a high DOL with minimal unspecific labeling is beneficial for fluorescence microscopy in general. Yet robust, versatile and easy-to-use tools for measuring cell-specific labeling efficiencies are not available. Here we present a DOL determination technique named protein-tag DOL (ProDOL), which enables fast quantification and optimization of protein-tag labeling. With ProDOL various factors affecting labeling efficiency, including substrate type, incubation time and concentration, as well as sample fixation and cell type can be easily assessed. We applied ProDOL to investigate how human immunodeficiency virus-1 pathogenesis factor Nef modulates CD4 T cell activation measuring total and activated copy numbers of the adapter protein SLP-76 in signaling microclusters. ProDOL proved to be a versatile and robust tool for labeling calibration, enabling determination of labeling efficiencies, optimization of strategies and quantification of protein stoichiometry.

Fluorescence microscopy has long been a vital tool in biological research, enabling the detection of proteins of interest (POIs) in a variety of contexts. However, measuring copy numbers of a POI reliably with fluorescence microscopy not only requires a methodology to count fluorophores, but also the information about the fraction of POI labeled with fluorescent markers. Affinity labeling techniques such as immunolabeling result in variable labeling efficiencies that are difficult to characterize. On the other hand, while genetic fusion with fluorescent proteins can yield a one-to-one ratio of label to the POI, it is often not suited for quantitative measurements and can be challenging due to insufficient photostability and ill-defined brightness states[1]. Additionally, fluorescent proteins in fusion constructs can exhibit a lower apparent labeling efficiency due to variable maturation efficiencies or inefficient chromophore formation in different subcellular environments[2–6]. Self-labeling protein tags, such as SNAP-tag[7,8] and HaloTag[9] that bind at most one label per tag are also genetically fused to the POI and can bind a variety of fluorescent substrates with potentially superior photophysical properties[1]. However, due to the

[1]Institute of Cardiovascular Sciences, College of Medical and Dental Sciences, University of Birmingham, Birmingham, UK. [2]School of Chemistry, College of Engineering and Physical Sciences, University of Birmingham, Birmingham, UK. [3]Centre of Membrane Proteins and Receptors, The Universities of Birmingham and Nottingham, Birmingham, UK. [4]Institute of Physical Chemistry, Heidelberg University, Heidelberg, Germany. [5]Department of Infectious Diseases, Integrative Virology, University Hospital Heidelberg, Heidelberg, Germany. [6]Division Systems Biology of Signal Transduction, German Cancer Research Center (DKFZ), Heidelberg, Germany. [7]Institute of Pharmacy and Molecular Biotechnology, Heidelberg University, Heidelberg, Germany. [8]Present address: Department of Pharmaceutical Chemistry, University of California, San Francisco, San Francisco, CA, USA. [9]Present address: PicoQuant GmbH, Berlin, Germany. [10]These authors contributed equally: Stanimir Asenov Tashev, Jonas Euchner, Klaus Yserentant. ✉e-mail: d.herten@bham.ac.uk

additional labeling step that requires incubation with fluorescent substrates, labeling efficiencies can vary depending on the chosen labeling condition. Moreover, unspecific binding of these substrates in the sample needs to be accounted for as it can result in unspecific signals affecting any subsequent quantitative or colocalization analysis.

To ensure optimal labeling conditions for the specific imaging technique, the degree of labeling (DOL) or the ratio of fluorescent markers to POI needs to be determined. A precisely determined DOL can also serve as a correction factor for measured protein copy numbers in complexes and protein concentrations obtained from fluorescence microscopy techniques. However, determining the DOL can be challenging[1,10,11] and different methods to address this issue have previously been developed[12]. One common approach is based on molecular counting standards such as the nuclear pore complex (NPC) combined with fluorophore counting methods such as super-resolution-based effective labeling efficiency (ELE)[10], quick photobleaching step analysis (quickPBSA)[1] or counting by photon statistics (CoPS)[13,14]. Unfortunately, using protein complexes with known stoichiometry often comes with substantial limitations, as complete complex assembly must be ensured and use of a homozygous knockin cell line is required. Additionally, methods such as ELE and CoPS require the use of labels suitable for super-resolution microscopy or specialized instrumentation, further limiting general application.

Colocalization analysis with an additional, spectrally different label in close spatial proximity has been proposed previously, using, for example, a fusion between SNAP-tag and HaloTag to estimate the DOL of both tags[15]. Other work has utilized an additional antibody labeling against the protein tags[16]. However, all proposed methods can suffer from unspecific labeling of the reference signal resulting in an underestimation of DOL.

To overcome these limitations, we propose a modular DOL calibration probe that employs a fluorescent protein as a nearly background-free reference signal combined with protein tags. This construct can be transiently or stably expressed in various cell lines and provides a way to measure labeling efficiency through colocalization at the single-molecule level, thus enhancing the reliability and versatility of the measurements. In the current implementation the DOL calibration probe is composed of a membrane anchored enhanced green fluorescent protein (eGFP) fused to a SNAP-tag and HaloTag and is named the 'protein-tag degree of labeling (ProDOL) probe'. Additionally, we developed a ProDOL analysis pipeline for labeling efficiency measurements by single-molecule colocalization analysis. The integration of these techniques greatly enhances our ability to characterize fluorescent labeling approaches and measure labeling efficiencies, thereby facilitating accurate counting of POIs. By identifying optimal labeling conditions for robust protein counting, we can better harness the potential of fluorescence microscopy in biological research, leading to a more accurate and reliable understanding of cellular processes. We demonstrate this potential by determining the time-resolved protein copy number of SLP-76 in microclusters (MCs) upon T cell receptor (TCR) activation and how these are affected by the human immunodeficiency virus (HIV)-1 protein Nef.

## Results

### ProDOL

ProDOL is based on the colocalization of the single-molecule signals emitted by labeled protein tags and reference labels in spectrally separated images. To compute the DOL, the fraction of reference label colocalized with a protein tag signal is assessed (Fig. 1b). We implemented this strategy by creating a fusion construct to serve as ProDOL probe with the ability to assess the labeling efficiency of different protein tags in mammalian cell lines (for Addgene ID, see Methods). The ProDOL probe was engineered with eGFP serving as a reference marker and two self-labeling protein tags (HaloTag and SNAP-tag). An N-terminal Lyn kinase-derived membrane anchor targets the ProDOL probe to the plasma membrane

via posttranslational modification to enable single-molecule imaging by total internal reflection fluorescence (TIRF) microscopy (Fig. 1a). An α-helical linker was added between SNAP-tag and HaloTag to facilitate maturation and avoid misfolding of the fusion protein, and a C-terminal His-tag allows affinity purification or immunolabeling (Fig. 1a).

With eGFP as the reference label, the ProDOL probe enables spectral discrimination of frequently used red and far-red fluorescent SNAP-tag and HaloTag substrates. If required, eGFP can be replaced by alternative fluorescent proteins to facilitate DOL measurements for protein-tag substrates with excitation or emission spectra overlapping with eGFP. In a similar fashion, alternative protein tags can be introduced into the ProDOL probe.

The expression level of the ProDOL probe is a critical parameter for robust DOL measurements. While high probe density provides better statistics, the density must be sufficiently low to reliably detect individual signals in diffraction-limited images. Therefore, the ProDOL probe was inserted into a retroviral pBABE plasmid to facilitate stable genomic integration and to achieve expression levels suitable for localization of diffraction-limited single-molecule signals[17].

Unspecific background signal is another important factor interfering with the determination of the DOL in living cells. While eGFP provides a background-free label, unspecific background has been frequently observed when labeling SNAP-tag or HaloTag in cells[18]. Therefore, we also created a truncated version of the ProDOL probe containing only the Lyn kinase membrane anchor and the eGFP reference label, but no additional protein tags (Lyn-eGFP (LynG)). The LynG probe serves as a control to enable monitoring the density of unspecific labeling (Supplementary Fig. 1).

To complement the probe, we created a data processing workflow consisting of seven modules accounting for the signal localization in cells, chromatic aberrations, unspecific signals and variations of the label densities (Fig. 1c). First, color multiplexed single-molecule images are acquired by TIRF microscopy (1), and a segmentation mask is generated from the reference channel image to exclude background signals outside of cells from subsequent analysis (2). The reference and label signals are localized with subpixel accuracy using ThunderSTORM[19] (3) and subsequently corrected for chromatic aberrations by applying an affine transformation matrix (4). Next, the colocalization of the labeled tags with the reference is determined (5) using a distance cutoff $T$ at which the fraction of specific colocalization is maximized while the contribution of random colocalization is kept at a minimum (Extended Data Fig. 1). The cutoff $T$ can in theory be set to the diffraction limit. However, nonlinear chromatic, spheric or planar aberrations can lead to a shift larger than the diffraction limit between spectrally dissimilar point spread functions (PSFs), thus making a variable cutoff $T$ beneficial. The colocalized signals are then adjusted to account for effects of the probe density on signal detection (6) (Extended Data Fig. 2). Multiple factors such as overlapping localizations, missed localizations, multiple assignment of signals or cutoff $T$ depend on the emitter density and can lead to an underestimation of the DOL. To obtain a robust result, DOL values determined in individual cells are averaged (7). To ensure that the established data processing routines result in unbiased DOL estimates, we validated the full pipeline using simulated data at ProDOL probe densities covering typically encountered experimental conditions (Extended Data Figs. 3 and 4).

### Cross-validation of ProDOL analysis

To experimentally validate the ProDOL method, we compared it with three approaches relying on defined protein complexes and different protein counting approaches. As calibration target for these experiments, we used nucleoporin 107 (Nup107), a key component of the NPC, present at 32 copies per complex[20]. Nup107-SNAP was expressed in a genome-edited U2-OS cell line from its native locus and was labeled with SNAP-Alexa Fluor 647 (SNAP-AF647)[21] following a shared staining protocol for all counting techniques (Fig. 2a). For protein counting, we

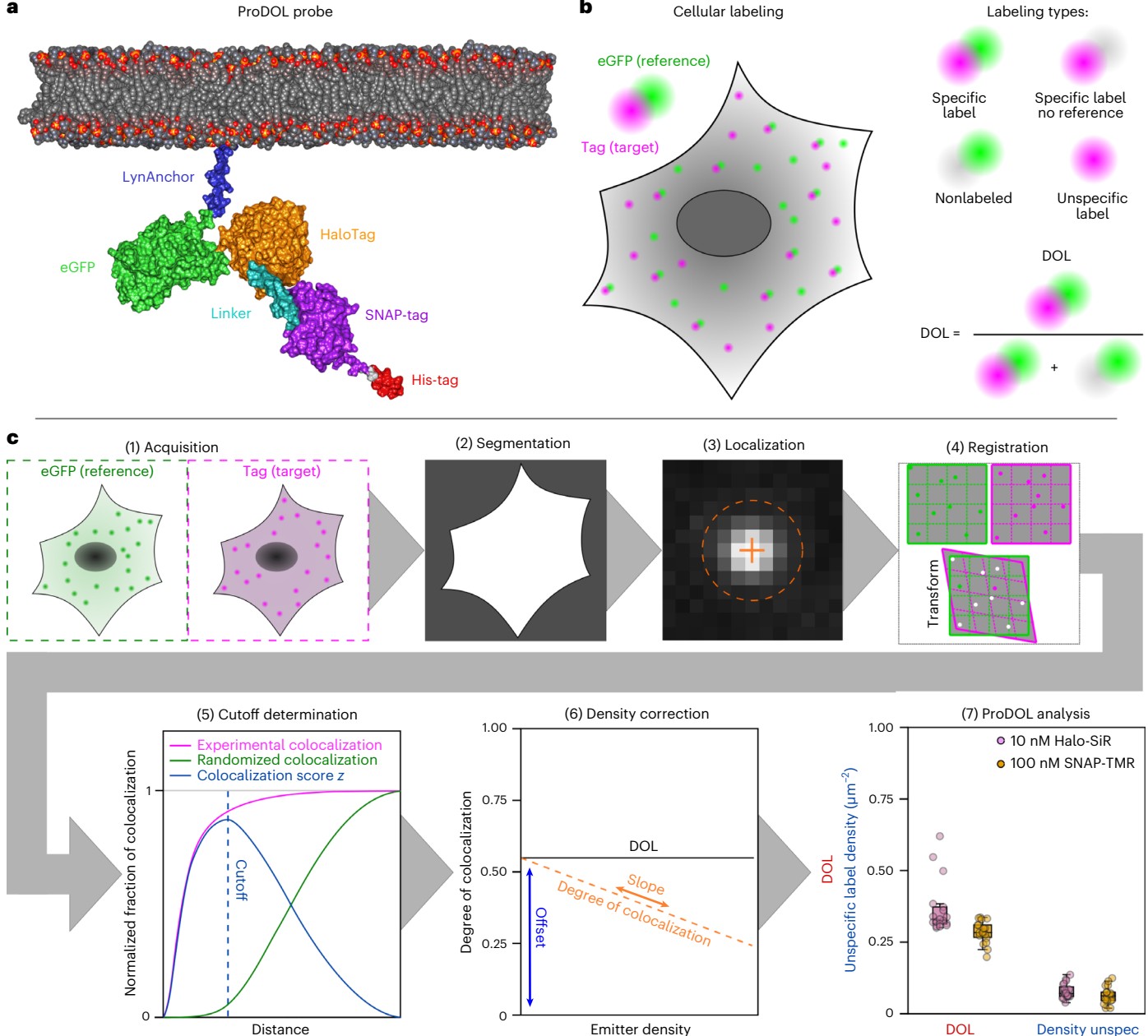

**Fig. 1 | Principle and workflow of ProDOL. a**, Molecular model of ProDOL probe generated using AlphaFold with postprediction modification using published structures of individual components (Protein Data Bank 2B3Q, 6Y8P and 6U32) and modeled lipid bilayer[35]. **b**, A schematic model to determine the DOL based on single-molecule colocalization analysis between an eGFP reference and tag-target label. Within the ProDOL analysis, four labeling types can be discerned (specific label, specific label with no reference, nonlabeled and unspecific label) and used to calculate the DOL and unspecific labeling density. **c**, ProDOL analysis workflow. (1) Acquired images having both reference and target channels are used as input. (2) The reference channel is used for generating a cell mask. (3) Reference and target signal localizations are fitted with subpixel accuracy. (4) The channels are aligned using affine registration of the localization data. (5) Calculation of colocalization cutoff *T*. (6) Factors are determined to correct the DOL for emitter density. (7) The DOL and the unspecific label density (Density unspec) are determined for all acquired tag-target channels. The box plots span the interval from the 25th to the 75th percentile with the median indicated by a horizontal line within the box. Whiskers extend to 1.5 × the interquartile range.

used three orthogonal emitter counting methods: ELE[10], quickPBSA[1] and CoPS[13,14]. ELE takes advantage of the known eightfold symmetry of the NPC to infer the DOL from analyzing single-molecule localizations within individual NPCs[9] (Fig. 2b). QuickPBSA detects stepwise intensity changes due to photobleaching during prolonged imaging to estimate the number of fluorophores[1] (Fig. 2c). CoPS is a quantum imaging technique estimating the number of emitters in diffraction-limited structures based on the detection of coincident photons[13] (Fig. 2d). In contrast to ELE, CoPS and quickPBSA do not require a priori knowledge of the underlying geometry or a specific fluorescent label.

Finally, ProDOL measurements were conducted in transiently transfected U2-OS cells expressing the ProDOL probe (Fig. 2e) using the same labeling conditions as for Nup107-SNAP described above. ProDOL analysis of the achieved labeling efficiency under these conditions revealed a DOL of 42.6 ± 5.3% (median ± s.d.). This is in agreement with the alternative quantification methods relying on the conserved copy number of Nup107/NPC where we determined labeling efficiencies of 42.2 ± 4.1%, 40.6 ± 5.8% and 40.5 ± 4.9% for ELE, quickPBSA and CoPS, respectively (Fig. 2f). Overall, a high degree of agreement across all four tested methods without significant differences (*P* = 0.330,

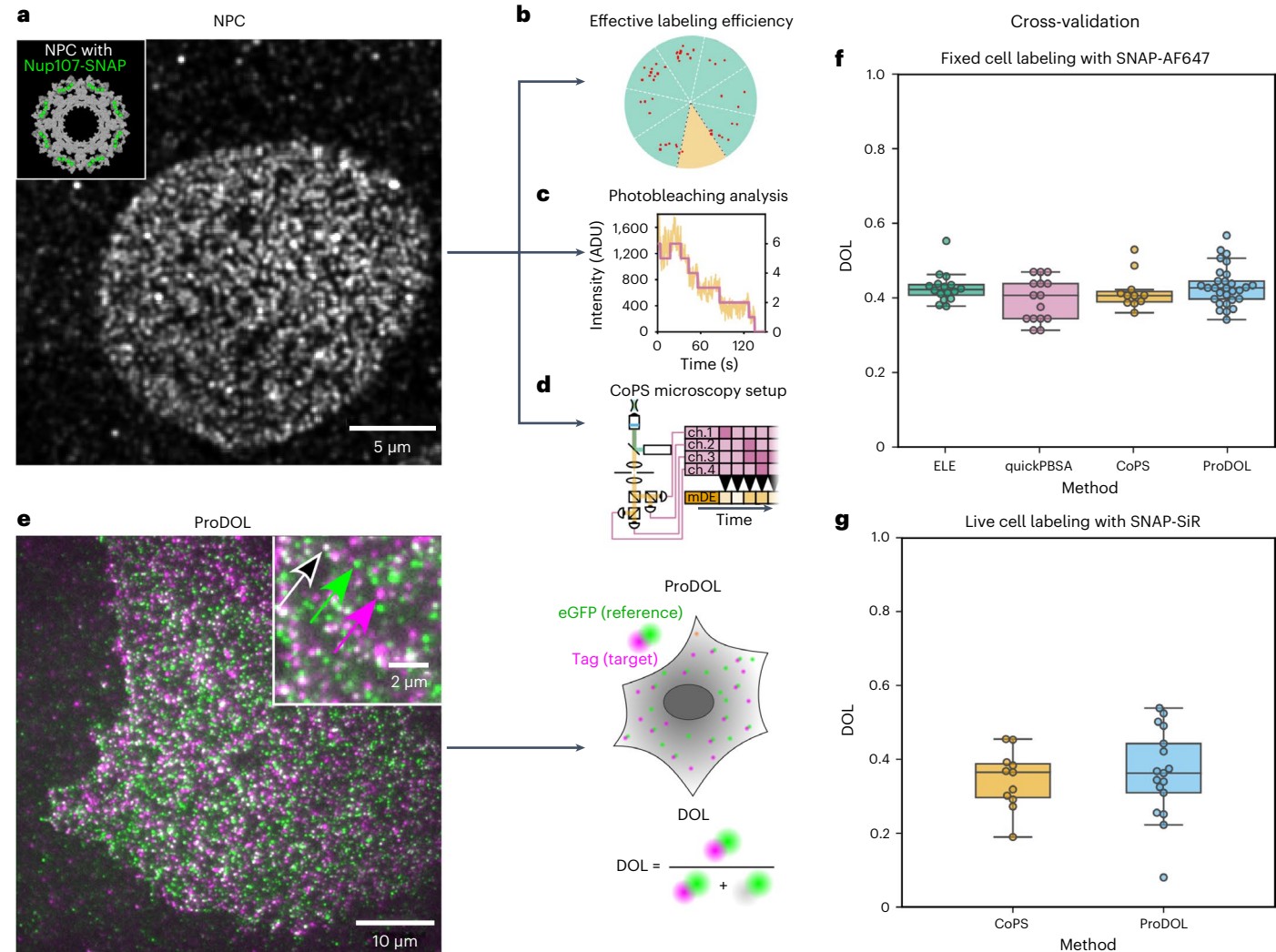

**Fig. 2 | Comparison of molecular quantification techniques and ProDOL.**
**a**, A representative TIRF image of SNAP-AF647 stained U2-OS cell with Nup107-SNAP knockin. Inset: a structural model of NPC (Protein Data Bank 5A9Q) highlighting the expected distribution of the 32 Nup107 (green) copies across the eight corners of NPCs. **b**, ELE analyzes the distribution of single-molecule localizations across individual NPCs and fits a binomial function to the number of occupied corners to estimate the DOL. **c**, PBSA relies on the stepwise reduction in fluorescence intensity (yellow) caused by photobleaching. Here Bayesian statistics are used to fit a change of states which correspond to the remaining emitting fluorophores (purple). **d**, CoPS microscopy setup with four single-photon avalanche detectors as independent detection channels (ch. 1–4) and a pulsed laser. A TCSPCS is used to generate a time trace of multiple detection

events (mDEs), the histogram of which is matched to a probability model to determine the number of emitters. **e**, TIRF image of a U2-OS cell transiently transfected with the ProDOL probe. The arrows depict single-molecule localizations for eGFP reference signal (green), SNAP-AF647 (magenta) and colocalization of both (white). **f**, A box and whisker plot of cell-wide DOLs after postfixation labeling by the indicated methods. In total, 16, 15, 11 and 29 cells were analyzed with ELE, quickPBSA, CoPS and ProDOL, respectively. **g**, A box and whisker plot of cell-wide DOLs after prefixation labeling determined by DOL ($N = 17$) and CoPS ($N = 11$ cells). Label distribution per NPC can be found in Extended Data Fig. 5. The box plots span the interval from the 25th to the 75th percentile with the median indicated by a horizontal line within the box. Whiskers extend to 1.5 × the interquartile range.

Kruskal–Wallis) between approaches demonstrates that ProDOL provides reliable DOL estimates without any specific requirements concerning the cell line and the fluorescent label and without requiring sophisticated emitter counting methods.

To remove the effects of fixation and permeabilization on our method, experiments were performed with SNAP-SiR in the same cell lines (Fig. 2g). ProDOL and CoPS analysis yielded labeling efficiencies of 36.1 ± 12.0% and 36.3 ± 7.9%, respectively, showing no significant difference between the two ($P = 0.707$).

**Measuring the degree of specific and unspecific labeling**

After successful validation, we utilized the ProDOL approach as a rapid screening tool to optimize labeling conditions by systematic variation of experimental parameters such as substrate concentration and

incubation time. Both pre- and postfixation labeling with SNAP-SiR and Halo-TMR in HeLa cells were characterized by inspection of DOL and unspecific labeling, with expression of ProDOL probe and LynG, respectively (Fig. 3a). As expected, a steady increase of the DOL for the two protein tags with increasing substrate concentrations (Fig. 3a, 1 and 3) both pre- (green) and postfixation (yellow) was observed showing saturation at substrate concentrations above 100 nM. When labeling prefixation, that is, in living cells, both substrates plateau before 40%, with the Halo-TMR reaching 35 ± 8% and the SNAP-SiR reaching 27 ± 4%. In comparison, postfixation labeling with SNAP-SiR showed a considerable increase in DOL (58 ± 7%), while the DOL achieved with Halo-TMR was only slightly higher at 40 ± 8%. In addition, the DOL achieved by the protein tags differs in their concentration dependence, in agreement with previously reported ligand affinities[22]. While labeling of HaloTag

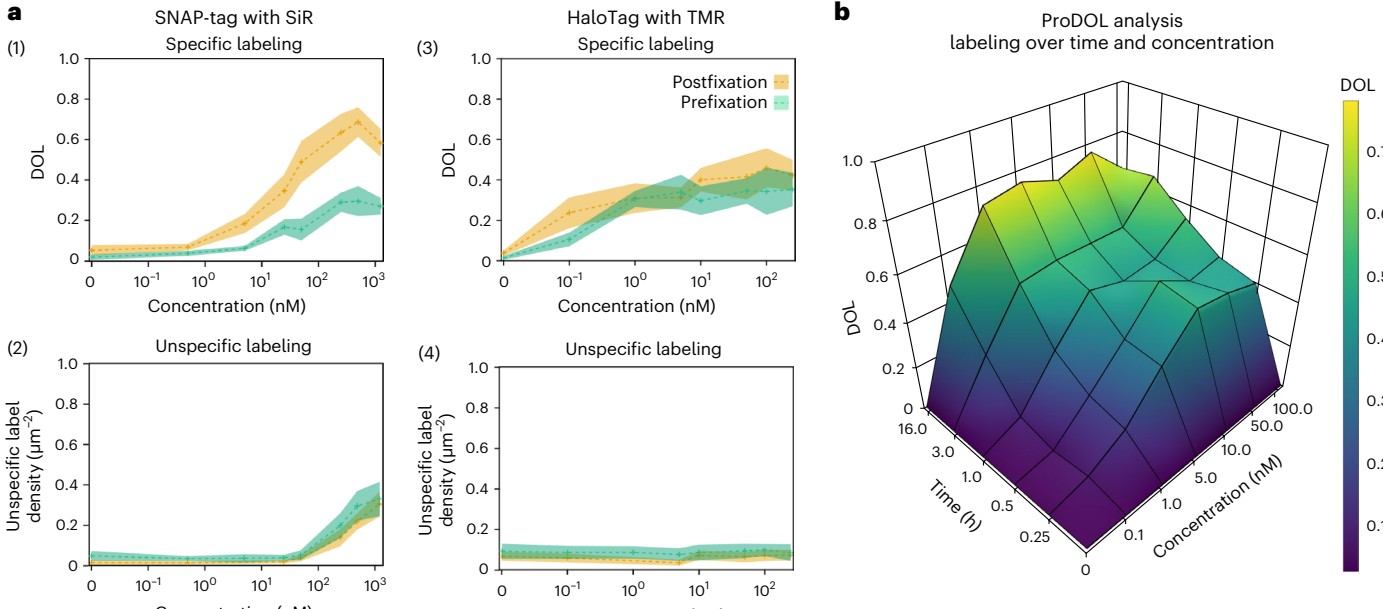

**Fig. 3 | Determination of DOL as a parameter of incubation time, dye concentration and labeling condition. a**, HeLa cells expressing ProDOL and LynG probes were stained with varying concentrations of SNAP-SiR (1, 2) and Halo-TMR (3, 4) for 30 min prefixation (green) or postfixation (orange). Determined DOL using ProDOL at varying substrate concentrations (1 and 3).

Unspecific labeling density measured in HeLa cells expressing LynG at varying substrate concentrations (2 and 4). **b**, DOL as a function of incubation time and Halo-SiR concentration in Jurkat T cells. In **a**, data are median ± s.d. (crosses, shaded bands) for 6–20 cells per condition. In **b**, data are median for 30–50 cells per condition.

can be achieved at subnanomolar substrate concentrations, we could only detect specific binding for SNAP-tag at concentrations above 1 nM. As this is the case under both pre- and postfixation labeling conditions, differences in the cell permeability of the dye substrates can be ruled out as reason for this behavior.

Knowledge of the DOL is essential for the quantification of protein copy numbers. In addition, unspecific labeling will also impact the measured protein copy numbers. To assess the density of unspecific labeling, the LynG probe was used in HeLa cells under identical labeling conditions as for the ProDOL probe and the unspecific labeling density was determined (Fig. 3a, 2 and 4). While we observed little difference between pre- and postfixation conditions, SNAP-SiR shows a significant increase in unspecific labeling beyond substrate concentrations of 100 nM, in contrast to Halo-TMR which shows less unspecific staining. Additional experiments with a range of different dye substrates in Huh7.5 cells indicate that this is probably due to their molecular properties, such as polarity and charge (Extended Data Fig. 6). Moreover, we cannot exclude effects associated with the cell type as we observed notable variations in the achieved DOL across different cell lines labeled with the same dye substrate (Extended Data Figs. 7 and 8).

To show the generalizability of the ProDOL approach for optimizing labeling conditions, we performed further experiments with Jurkat T cells (Fig. 3b) as well as Huh7.4 and H838 cells (Extended Data Figs. 7 and 8) expressing ProDOL labeled with Halo-TMR and SNAP-SiR. As expected, the DOL increases with incubation time and substrate concentrations. In combination with the associated quantification of unspecific labeling (Extended Data Figs. 7c,d, 8c,d and 9), this allows the selection of optimal labeling conditions depending on the experimental method and target.

### Quantifying the disruption of T cell signaling by HIV-1 Nef

To demonstrate how ProDOL can be applied to reveal protein complex stoichiometries in situ, we studied the mechanism of action of the HIV-1 pathogenesis factor Nef on TCR MCs. It is known that Nef alters the response of infected CD4 T cells of TCR activation[23,24] and that TCR

engagement triggers the formation of signaling competent protein MCs. The composition and activity of the MCs in turn determine the magnitude and breadth of signaling outputs. Further, Nef has been shown to affect TCR signaling by reducing the formation of MCs that contain the adapter protein SLP-76 (ref. 25). However, the quantitative information on the protein composition and activity of individual MCs upon infection of CD4 T cells with HIV-1 remains unknown[26]. To address this, we first used ProDOL in combination with CoPS to determine the absolute copy number of SLP-76 in individual MCs in response to TCR engagement in the absence or presence of Nef. Jurkat CD4 T cells stably expressing SLP-76-HaloTag were labeled in suspension with Halo-SiR and activated upon binding to glass coverslips functionalized with α-CD3 antibodies[27–29]. Cells in which individual SLP-76 clusters could be optically resolved were examined with CoPS 1.5, 5 and 10 min after activation (Fig. 4). In parallel, the DOL for the applied labeling conditions was determined in Jurkat T cells stably expressing the ProDOL probe to enable extrapolation of the labeled molecule numbers to absolute protein copy number of SLP-76-HaloTag in MCs (Fig. 4a). These measurements indicated that the number of SLP-76-HaloTag in MCs remained constant at around 20 protein copies per cluster between 1.5 and 10 min of activation (P = 0.7495). Interestingly, the associated label number distributions determined at these times show a log-normal distribution of increased variance (Fig. 4b and Supplementary Fig. 2), clearly indicating a lack of defined cluster organization, which can be explained by the multiplicative product of independent variables in accordance with the central limit theorem in log-space commonly found in intracellular distributions[30]. To assess the impact of HIV-1 Nef on MC composition, we transiently expressed an HIV-1 Nef-eGFP fusion protein or an eGFP control in Jurkat CD4 T cells stably expressing SLP-76-HaloTag. In addition to the expected reduction in the overall number of SLP-76 MCs[26], evaluating the stoichiometry of SLP-76 MCs revealed a significant reduction of SLP-76-HaloTag copies to 10.8 ± 2.2 per MC 5 min post activation (P < 0.0001; Fig. 4b).

Having shown that the overall abundance of SLP-76 in MCs remains constant in the tested 1.5–10-min activation window and that HIV-1

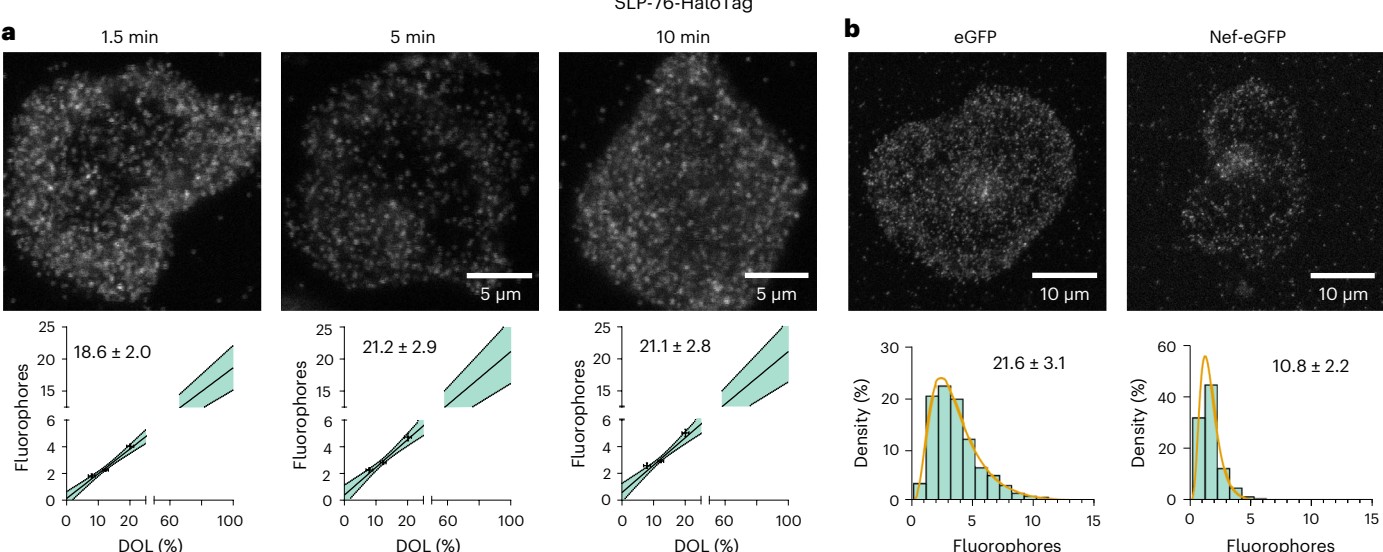

**Fig. 4 | SLP-76 stoichiometry in activated Jurkat CD4 T cells.** Jurkat CD4 T cells expressing SLP-76-HaloTag stained with Halo-SiR. T cell activation of individual cells was synchronized by observing only cells that adhered to the coverslip within the first 3 s and all other cells washed away. **a**, Top: representative confocal images of SLP-76 clusters at different times after activation (1.5 min, 5 min and 10 min). Bottom: extrapolation of the copy number of SLP-76-HaloTag per cluster using different DOLs (8%, 12% and 20%). **b**, Top: representative confocal images of SLP-76 clustering 5 min after activation in Jurkat CD4 T cells transfected with eGFP (control) or Nef-eGFP. Bottom: histograms of labeled SLP-76-Halo per cluster (at DOL of 16.7 ± 1.7%) in the presence and absence of the viral protein

Nef. In **a**, DOL 8%: $n$ = 21 cells; DOL 12%: $n$ = 20 cells; and DOL 20%: $n$ = 9 cells. Cluster analysis for 8% DOL 1.5 min: $n$ = 110 clusters; 5 min: $n$ = 268 clusters; and 10 min: $n$ = 72 clusters. For 12% DOL 1.5 min: $n$ = 59 clusters; 5 min: $n$ = 175 clusters; and 10 min: $n$ = 41 clusters. For 20% DOL 1.5 min: $n$ = 65 clusters; 5 min: $n$ = 237 clusters; and 10 min: $n$ = 41 clusters. Error bars of data points represent DOL: mean ± s.e.m.; cluster analysis: geometric mean of log-normal fit ± s.e.m.; black line: best fit of linear regression; shaded region: 95% confidence interval of linear fit. In **b**, DOL analysis: $n$ = 24 cells; cluster analysis for eGFP: $n$ = 1348 clusters; Nef-eGFP: $n$ = 380 clusters; and orange line: best fit of log-normal distribution.

Nef reduces the SLP-76 copy number per MC, we next determined the number of active SLP-76 copies per MC (Supplementary Note 1). For this, we determined the phosphorylation state of SLP-76 in MCs via semi-quantitative immunolabeling with fluorescently labeled primary antibodies (Supplementary Fig. 3) in combination with CoPS (Supplementary Fig. 2). These experiments revealed that phosphorylation of SLP-76 increases over the examined time, unlike the copy numbers of SLP-76-Halo. In contrast, the presence of HIV-1 Nef decreased both the copy numbers and phosphorylation of SLP-76. Together, these results indicate that the function of SLP-76 in early TCR signaling is regulated by phosphorylation in individual MCs, but the disruption of its function in the presence of pathogenesis factor Nef is tied to the presence of it in MCs.

## Discussion

The ProDOL probe in combination with the established data analysis pipeline provides a robust and versatile labeling calibration workflow. Unlike other approaches, ProDOL allows for determination of the DOL across a variety of mammalian cell lines with potential use in other species where transient expression of a transgenic construct is possible. Additionally, ProDOL can be carried out on any single-molecule sensitive microscopy setup with diffraction-limited acquisition, as no super-resolved acquisition is needed, expanding both on the feasibility, and dye compatibility. We have further validated the robustness of ProDOL in determining labeling efficiencies and its sensitivity to small methodological changes with extensive experiments and simulations (Extended Data Fig. 3). We therefore conclude that ProDOL can be used to determine optimal staining conditions for protein-tag labeling for the specific target and microscopy method, as well as to measure the DOL of proteins in the cytoplasm of cells. The fact that achieved labeling efficiencies and unspecific background staining depend on the chosen labeling parameters including, but not limited to substrate type, concentration and incubation time, strongly suggests

that whenever possible, labeling efficiency calibration measurements should be performed in conjunction with labeling of the POI. If such simultaneous calibration measurements are not feasible, one way to achieve reproducible labeling efficiencies across multiple experiments is to perform comprehensive labeling efficiency screens, as shown in Fig. 3b and Extended Data Figs. 7, 8 and 9 to identify parameter spaces that result in stable labeling efficiencies even upon small variations in labeling conditions.

Determination of DOL assumes similar labeling efficiencies between the POI and the DOL probe. While this is not guaranteed, we provide evidence that similar DOLs are achieved for the plasma membrane-localized ProDOL probe and for NPCs. In addition, this also shows that comparable labeling efficiencies are achieved for protein tags sparsely distributed across the plasma membrane as well as for nucleoporins, which are known to form high density structures embedded in the nuclear membrane. However, additional subcellular compartments such as the lumen of intracellular organelles or the extracellular leaflet of the plasma membrane warrant further studies to systematically dissect the influence of subcellular protein tag localization on achieved DOL.

ProDOL highlights the need for robust and easy to implement labeling calibration as most, if not all, labeling strategies result in substoichiometric labeling[10]. Interestingly, paraformaldehyde (PFA) fixation before labeling increased the maximal labeling efficiency achieved for SNAP-tag labeling, while HaloTag labeling was not notably affected.

The ProDOL probe can be easily modified to accommodate alternative peptide or protein tags such as CLIP, TMP-tag or HA-tag[31–33] and is in principle also suited to determine the maturation efficiency of fluorescent proteins. A comprehensive comparison of different tags with the ProDOL pipeline will help to identify systems that achieve high labeling efficiency and minimum background staining, which will be especially useful for super-resolution imaging as a high labeling

density is crucial for appropriate sampling of structures at highest resolution. The ProDOL concept can also be applied to determine labeling efficiencies in alternative model organisms and, in principle, also for immunolabeled samples. In its current form, the ProDOL probe is potentially suitable for measuring labeling efficiencies of commonly used anti-GFP and anti-His-tag immunolabels, but incorporation of additional epitopes for which immunolabels are available can extend its applicability beyond these targets. However, the use of immunolabels requires additional calibration of the number of fluorophores conjugated to individual immunolabels, and additional controls to ensure comparable immunolabel binding to the modified ProDOL probe and target structures may be required. It has recently been shown that a colocalization-based DOL determination approach similar to the ProDOL workflow in combination with DNA points accumulation for imaging in nanoscale topography (DNA-PAINT) can be used to partially address the challenges associated with immunolabeling of target molecules[34]. However, indirect immunolabeling poses an additional challenge because the use of a secondary labeling reagent will lead to additional off-target binding and to a broadening of the label number distribution per target molecule.

By applying the ProDOL concept to T cell activation, we showed how the composition of signaling MCs changes over time. We were able to demonstrate that the copy number of SLP-76 per MC stays constant between 1.5 and 10 min after activation. Thus, SLP-76 is either recruited in an early phase of T cell activation not covered by these experiments or already pre-assembled in MCs before activation. We also applied ProDOL to study how the HIV-1 pathogenesis factor Nef rewires TCR signaling to the benefit of the virus. Here we established for SLP-76 that the viral protein primarily acts to prevent recruitment of host cell components into signaling hubs. Supplementary experiments showed that the fraction of phosphorylated SLP-76 increases during activation and that the phosphorylation status of SLP-76 in successfully assembled MCs is not modulated by Nef (Supplementary Fig. 2 and Supplementary Note 1). ProDOL will now enable analogous analysis for more components of the TCR signaling machinery and profoundly impact the development of strategies to therapeutically restore physiological TCR signaling in HIV-1-infected CD4 T cells.

Taken together, ProDOL will be a valuable tool for various use cases, including but not limited to label optimization, quantitative super-resolution microscopy, protein complex stoichiometry determination and as quality control run along experiments for validating labeling protocols. A simple and modifiable probe design and accessible data analysis routine enables the community to develop upon the concept of determining the DOL using colocalization analysis. We anticipate that the broad applicability of the ProDOL workflow will enable routine measurements of labeling efficiencies extending the use of quantitative fluorescence microscopy for measuring absolute protein copy numbers in cell biology and beyond.

## Online content

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

## Methods

### Cloning

A ProDOL precursor was synthesized as gBlock (Integrated DNA Technologies) consisting of Lyn kinase membrane anchor (N-terminal amino acid: 1–13), α-helical linker[36], SNAP-tag (SNAP26m) and His-tag. Between all segments, unique restriction sites were included allowing for easy modification by restriction/ligation cloning (Supplementary Fig. 1). The precursor was inserted into pMOWS vector using EcoRI, PacI restriction sites[37].

eGFP and HaloTag were inserted between the LynAnchor and linker using the unique restriction sites XhoI, BamHI and AgeI. Next, the SNAP26m-tag was exchanged to a SNAP$_f$-tag using NdeI and MfeI restriction sites. Finally, the full ProDOL probe was cloned into the pBABE expression vector.

For generating the LynG construct, pBABE-ProDOL was digested using BamHI and MfeI, leaving only the LynAnchor and eGFP. After blunting the ends, the plasmid was religated, generating pBABE-LynG. Successful integration was confirmed for each vector using restriction digestion.

For generating stable Jurkat CD4 T cell lines, ProDOL and LynG constructs were transferred into a pWPI vector. For this, both constructs were PCR amplified from the respective pBABE vector with addition of PmeI and SpeI restriction sites during PCR. PCR products were ligated into the pWPI vector. Successful integration was confirmed for each vector using restriction digestion.

Full sequences for ProDOL and LyNG probes are given in Supplementary Table 1. Plasmids are available through Addgene (pBABE-ProDOL probe: #206866 and pBABE-LynG probe: #206867).

### Cell culture

Adherent mammalian cells were cultured in a complete growth medium consisting of phenol red-free Dulbecco's modified Eagle medium (DMEM), supplemented with 10% fetal bovine serum (FBS), 1× GlutaMAX and 1 mM sodium pyruvate (all Gibco). Cells were incubated at 37 °C, 5% $CO_2$ and 100% humidity. Adherent cells stably expressing ProDOL or LynG probes were additionally treated with 1.5 µg ml$^{-1}$ puromycin to maintain transgene expression. Genetically modified cells were additionally treated with the appropriate selection antibiotics. Upon reaching approximately 80% confluency, cells were subcultured. The protocol involved aspiration of growth medium, a single rinse with PBS and subsequent incubation with TrypLE Express until cells fully detached. Inactivation of TrypLE was achieved via the addition of twice the volume of complete growth medium, followed by centrifugation at 500g for 5 min to pellet the cells. After removal of the supernatant, the cell pellet was resuspended in complete growth medium and subsequently seeded at dilution ratios ranging from 1:6 to 1:10.

Suspension cells were cultured in complete growth medium consisting of phenol red-free RPMI-1640 medium, 10% FBS, 1× GlutaMAX, 1 mM sodium pyruvate and 1× penicillin–streptomycin (Gibco). Cells were incubated at 37 °C, 5% $CO_2$ and 100% humidity. Genetically modified cells were additionally selected by the appropriate selection antibiotics. All suspension cells were passaged into fresh medium to maintain a concentration of between $2 × 10^5$ and $8 × 10^5$ cells ml$^{-1}$.

### Stable cell line generation

Stable adherent transgenic cell lines, expressing ProDOL or LynG probes, were established through the Phoenix-Ampho retroviral transduction system utilizing pBABE-ProDOL and pBABE-LynAnchor-eGFP (LynG) plasmids. These plasmids were introduced into the Phoenix-Ampho virus packaging cell line via calcium phosphate precipitation for 6 h, resulting in the formation of replication-deficient viral particles for subsequent transduction. Collected virus was collected 24 h posttransfection after passing through 0.22-µm syringe filters.

The transduction of Huh7.5 (ref. 38), H838 (NCI-H838, American Type Culture Collection) or HeLa (American Type Culture Collection)

cells was achieved through spin transduction at 340g for 3 hours. The selection process of transduced cells with puromycin was initiated 24 h after transduction. For single-molecule imaging, Huh7.5 and HeLa ProDOL and LynG cell lines were sorted by fluorescence-activated cell sorting (FACS) to yield cell lines with suitable expression levels.

Stable transgenic Jurkat CD4 T cell lines, expressing ProDOL or LynG, were established using a second-generation lentiviral system. HEK293T cells were transfected with lentiviral vectors VSV-G and PAX2 in combination with a pWPI vector (pWPI-ProDOL or pWPI-LynAnchor-eGFP (LynG) plasmid) at a 1:2:3 ratio using polyethyleneimine (1 µg µl$^{-1}$). At 48 h posttransfection, supernatant was collected and filtered using 0.48-µm filters.

Then, $3 × 10^6$ Jurkat CD4 T cells were resuspended in 1 ml viral supernatant and centrifuged at 1,000g for 90 min, and 5 ml medium was added 16 h post transduction. For single-molecule imaging, Jurkat CD4 T cells were sorted through FACS to yield cell lines with suitable expression levels.

### FACS

FACS was performed on an Aria III cell sorter using FACSDiva version 8.0.1 (BD Biosciences). The gating of live cells was carried out based on the front and side scatter signal. Further sorting was conducted based on the fluorescence signal generated from 488 or 633 nm excitation for eGFP- and 1 nM Halo-SiR-labeled samples respectively. The sorted cells were gathered into tubes filled with prewarmed complete growth medium supplemented with 100 U ml$^{-1}$ penicillin and 100 µg ml$^{-1}$ streptomycin. Expression of full-length ProDOL and LynG probes by the established cell lines was subsequently confirmed by western blot.

### Western blot

To examine the Nef-eGFP and eGFP expression, cells were lysed in RIPA buffer while mixing on a rotator for 30 min before sonication on ice. The samples were then centrifuged at 20,000g for 10 min and the supernatant was collected. Sample buffer (64 g l$^{-1}$ SDS, 40 mM Tris pH 7.4, 8% glycerol, 12.3 g l$^{-1}$ dithiothreitol, 0.16 g l$^{-1}$ bromophenol blue and 20% 2-mercaptoethanol) was added and the mixture was heated for 2 min at 95 °C. Samples were run on a 10% SDS polyacrylamide gel before blotting on a polyvinylidene fluoride membrane in Laemmli buffer for 1 h at 260 mA. Membranes were blocked in 5% bovine serum albumin (BSA) in Tris-buffered saline 0.2% Tween-20 (TBST) before incubating overnight at 4 °C with primary anti-GFP (1:1,000 dilution) in 1% BSA in TBST. After washing with TBST, the membranes were incubated for 1 h with secondary antibodies (1:10,000 dilution) conjugated with horseradish peroxidase at room temperature. Membranes were washed with TBST and developed in a 1:1 mixture of 0.1 M Tris at pH 8.5, 1.1 mg l$^{-1}$ luminol, 0.185 mg l$^{-1}$ p-coumaric acid, 1% dimethylsulfoxide and 0.018% $H_2O_2$.

### Cell seeding

Lab-Tek eight-well chambered imaging slides were cleaned by incubating twice with 0.1 M hydrofluoric acid for 30–60 s followed by washing twice with water. Thereafter Lab-Teks were incubated with PBS for at least 5 min. For adherent cell lines, chambers were used without further modification at a concentration of $1 × 10^4$ cells cm$^{-2}$.

For T cell experiments, 200 µl of 0.01% poly-L-lysine (Sigma-Aldrich) solution was added to eight-well Lab-Tek chambered imaging slides and incubated for 10 min before removing solution and letting it air dry. Subsequently, the chambers were incubated for 3 h at 37 °C with 100 µl of α-CD3-antibody (BioLegend, V CD03.05) solution (10 µg ml$^{-1}$ in PBS). Chambers were then washed three times with PBS before seeding of cells.

For ProDOL measurements, $2 × 10^4$ cells in 100 µl PBS were added to each chamber post labeling and incubated for 10 min before fixation with freshly prepared 4% PFA. For time-resolved activation studies similar to but quicker than in previously published protocols, $2 × 10^6$ cells in

100 µl PBS were added to each chamber and after 3 s fully aspirated[26]. After full aspiration, 100 µl PBS was gently added and incubated for 1.5 min 5 min, or 10 min before fixation with 4% PFA.

### Transfection

Transient transfection of adherent cells was carried out using FuGENE HD. First, 0.2 µg DNA and 0.6 µl FuGENE HD were added to 10 µl Opti-MEM. The mixture was incubated for 15 min at room temperature before adding to the cells in a Lab-Tek eight-well chamber slide. At 24 h posttransfection, cells were used for further analysis.

Jurkat CD4 T cells were transiently transfected with eGFP or Nef-eGFP expression plasmids by electroporation. First, $5 \times 10^6$ cells were collected by centrifugation, washed with 10 ml of serum-free medium, resuspended in 500 µl serum-free medium and transferred into a 0.4-cm electroporation vial. Then, 20 µg of respective plasmid DNA was added and the electroporation was performed at 250 V and 950 µF for 21 ms using a Gene Pulser XCell electroporation device. Afterward, the cells were transferred to a fresh six-well chamber with 5 ml complete growth medium. At 24 h posttransfection, cells were used for further analysis.

### Labeling protocols

**Pre- and postfixation labeling of adherent cells.** Cells expressing ProDOL probe or LynG were stained with 0.1–1,000 nM of Halo-dye or SNAP-dye conjugate for 0.25–16 h in live or fixed cells. Cells were then washed four times with prewarmed DMEM at 37 °C and 5% $CO_2$ for 15 min, 15 min, 45 min and 5 min. Cells were fixed with freshly prepared and prewarmed electron microscopy-grade 3.7% PFA (EMS) in PBS for 40 min. All cells were imaged in serum-free medium.

**NPC labeling in genome-edited U2-OS Nup107-SNAP cells.** For cross-validation analysis with postfixation labeling, U2-OS cells were fixed with freshly prepared 2.4% PFA for 30 s before permeabilization with 0.4% (v/v) Triton X-100 in PBS, followed by fixation again in 2.4% PFA. Samples were quenched with 100 mM $NH_4Cl$ in PBS and washed twice for 5 min before a 30 min incubation in an Image-iT Signal Enhancer. Samples were stained with 1 µM SNAP-AF647 diluted in 1 µM dithiothreitol, and 0.5% BSA in PBS for 2 h. The samples were washed three times in PBS for 5 min before imaging.

For live cell staining, U2-OS cells were processed as described in ref. 1. Briefly, the cells were incubated for 2 h with 100 nM SNAP-SiR in complete DMEM, after which the cells were washed in DMEM twice quickly, then twice for 30 min and once for 1 h. Then, cells were fixed in 3.7% PFA and washed twice with PBS.

**Jurkat CD4 T cell HaloTag staining.** For live cell staining, Jurkat CD4 T cells were collected by centrifugation for 3 min at 200$g$, after which the medium was replaced with full growth RPMI-1640 medium containing 0.1 nM to 100 nM Halo-SiR for 15 min to 16 h at 37 °C (Fig. 3b and Extended Data Fig. 9), 10 nM for variable durations (Fig. 4a) or 1 nM for 1 h (Fig. 4c). The cells were then washed by spinning the sample down and replacing the staining solution with fresh media after 15 min three times and then a final 40 min wash in RPMI. The cells were spun down one final time before being resuspended in PBS.

**Phospho-SLP-76 immunofluorescence.** The antibody (Abcam, ab206782) was labeled with SiR at a DOL of 1.27 ± 0.05 as measured by absorption spectroscopy to approximate a 1:1 stoichiometry of label versus antibody. Determining the label number distribution of the labeled antibodies using CoPS (Supplementary Fig. 3) enabled a more quantitative assessment of the phosphorylation during T cell activation.

Antibody staining of Jurkat CD4 T cells was conducted on cells already fixed on the coated coverslips. The sample was permeabilized with 0.1% Triton X-100 (Sigma-Aldrich) with 1 mM $Na_3VO_4$ in PBS for

5 min before being washed three times with 1 mM $Na_3VO_4$ in PBS. The sample was blocked with PBS containing 5% FBS and 1 mM $Na_3VO_4$ for 30 min before incubating overnight with 2 µg ml$^{-1}$ anti-pY145 SLP-76. Finally, the sample was washed three times with PBS before postfixation with 2% PFA in PBS for 15 min.

### Microscopy setup

**CoPS.** CoPS measurements were performed on two custom-built confocal microscopes.

Microscope I (for data shown in Fig. 4 and Extended Data Figs. 4, 7 and 8) (Axiovert 100, Zeiss) was equipped for sample-scanning confocal optical microscopy. The microscope setup included a $XY$ and $Z$ piezostage (Physik Instrumente) for nanometer-resolution positioning and an AlphaPlan-Fluar 100×/1.45 oil immersion objective (Zeiss). The microscope included a <90 ps-pulsed laser diode emitting at 640 nm (LDH P-C-640B, PicoQuant, 20 MHz repetition rate) and four single-photon sensitive avalanche photodiodes (APDs) (Perkin-Elmer). The excitation lasers were coupled into a single-mode polarization maintaining fiber (Schäfter+Kirchhof). A dichroic mirror (z532/640, CHROMA) was employed to separate the paths of the emission and excitation beams. The emitted signal was filtered using a quadband notch filter with additional spatial filtering using a 100-µm pinhole placed in the focal plane between two achromatic doublet lenses. All emission was split into four equal intensity paths using 50:50 beamsplitters and focused on the four APDs with 685/70 nm bandpass filters placed in front of each APD. Signals detected by the APDs were processed using a multichannel time-correlated single-photon counting system (TCSPCS) (HydraHarp400, PicoQuant). The microscope was operated using SymPhoTime 64. The exposure settings and the illumination intensities were tuned for each sample.

Microscope II (for data shown in Fig. 2 and Extended Data Figs. 5 and 9) (Eclipse Ti; Nikon) was equipped for laser-scanning confocal optical microscopy (Flimbee, PicoQuant). The microscope setup included a motorized $XY$ stage (Marzhäuser), a PFS2 autofocus system and an Apo TIRF 100× 1.49 NA oil immersion objective (Nikon). The microscope included a <90 ps-pulsed laser diode emitting at 640 nm (LDH P-C-640B, PicoQuant, 20 MHz repetition rate) and four APDs (Perkin-Elmer). The excitation lasers were coupled into a single-mode polarization maintaining fiber (Schäfter+Kirchhof). A dichroic mirror (z532/640, CHROMA) was employed to separate the paths of the emission and excitation beams. The emitted signal was filtered using a quadband notch filter with additional spatial filtering using a 100-µm pinhole placed in the focal plane between two achromatic doublet lenses. All emission was split into four equal intensity paths using 50:50 beamsplitters and focused on the four APDs with 685/70 nm bandpass filters placed in front of each APD. Signals detected by the APDs were processed using a multichannel TCSPCS (HydraHarp400, PicoQuant). The microscope was operated using SymPhoTime 64. The exposure settings, and the illumination intensities were each tuned for each individual sample.

**TIRF microscope.** ProDOL and quickPBSA measurements were carried out on a custom-built widefield microscope (Eclipse Ti, Nikon) equipped with both epifluorescence and TIRF illumination. The microscope setup included a motorized $XY$ stage (Marzhäuser), a PFS2 autofocus system and an Apo TIRF 100× 1.49 NA oil immersion objective (both Nikon). Images were captured using an iXon Ultra 897 back-illuminated emCCD camera (Andor). The microscope utilized a fiber-coupled multilaser engine (Toptica Photonics), equipped with 405, 488, 561 and 640 nm laser lines for illumination. A quadband dichroic mirror was employed to separate the paths of the emission and excitation beams. The emitted signal was filtered using a quadband notch filter with additional bandpass filters (525/50, 605/70 and 690/70 nm), which were installed in a motorized filter wheel (Thorlabs) between body and an OptoSplit II (CAIRN). The microscope was operated using µManager

1.4 (ref. 39). The exposure times, the electron-multiplying gain and the illumination intensities were each adjusted for each sample.

**dSTORM super-resolution microscope.** ELE measurements were carried out on a custom-built widefield microscope (RAMM, ASI) equipped with both epifluorescence and TIRF illumination. The microscope setup included a motorized *XY* stage (ASI), a CRISP auto-focus (ASI) and an Apo TIRF 100× 1.49 NA oil immersion objective (Nikon). Images were captured using a Prime95B scientific CMOS (sCMOS) camera (Photometrics) with 130 × 130 µm field of view at 111.5 nm px$^{-1}$. The microscope utilized four laser lines for illumination: 405 nm (iBEAM-smart-405, Toptica), 488 nm (Cyan Laser-40, Spectra Physics), 561 nm (gem 561, Laser Quantum) and 647 nm (2RU-VFL-P-2000-647, MPBC) controlled by two AOTFnC-Vis-TN (AA Opto-electronic) and passed through a beam shaper (piShaper, AdlOptica) for homogeneous illumination. A quadband dichroic mirror was employed to separate the paths of the emission and excitation beams. The emitted signal was filtered using a quadband notch filter with additional bandpass filters (450/40, 525/50, 593/46 and 731/137 nm), which were installed in a motorized filter slider (Thorlabs) between the microscope body and camera. The microscope was operated using µManager 2.0 (ref. 39) extended with custom microcontroller boards (Arduino). Exposure times, gain and illumination intensities were adjusted for each sample.

## Image acquisition
**CoPS.** Overview images were generated for each cell by confocal scanning. Protein clusters were localized using a custom-written analysis routine to detect local intensity maxima in acquired images. Spatially isolated maxima with a minimum distance of 500 nm to the next maxima position were selected for CoPS data acquisition. A total of $1 \times 10^7$ laser cycles (0.5 s) of photon coincidence events were recorded per cluster at 10 µW excitation power (measured before objective).

**quickPBSA.** Photobleaching step analysis (PBSA) was performed in ROXS PCD buffer[1]. Traces for U2-OS Nup107-SNAP cells were acquired at an irradiance of 1.2 kW cm$^{-2}$ at 640 nm. In total, 3,000 frames with an exposure time of 200 ms per frame were recorded for each acquisition.

**ELE.** NPCs in dSTORM were acquired at 35 ms exposure time, 20,000 frames and HiLo illumination, at ~4 kW cm$^{-2}$ irradiance at the sample plane. The 405 nm laser irradiance was gradually increased over time using a predetermined exponential function (maximum 30 W cm$^{-2}$), and 1 ml of 35 mM mercaptoethylamine glucose oxidase–catalase buffer according to ref. 40 was used in a custom-made airtight sample holder.

**ProDOL.** ProDOL measurements were acquired at 50 ms exposure time, 10.6 mW (640 nm), 3.4 mW (561 nm) and 6.9 mW (488 nm) laser power (measured before the objective) and a gain of 100. Ten to twenty frames were recorded and averaged per measurement. Images were acquired beginning with 640 nm excitation, followed by 561 nm excitation and finally 488 nm excitation. Then 2 min of photobleaching at 561 nm (17 mW) was carried out before acquisition of the 488 nm excitation channel to reduce the potential of Förster energy transfer between eGFP and red-shifted fluorophores excited at 561 nm.

**Antibody DOL calibration.** Phospho-SLP-76 antibodies were labeled with SiR-NHS (Spirochrome) and purified using Zeba spin desalting columns (40 K molecular weight cutoff, Thermo Fisher Scientific). DOL analysis was performed according to equation (1) and yielded an ensemble DOL of 1.27 at 65% protein yield after purification.

$$\text{DOL} = \frac{\text{Abs}_{652} \times \varepsilon_{\text{Target}}}{\varepsilon_{\text{Label}} \times (\text{Abs}_{280} - \text{Correction}_{280}\text{Abs}_{652})} \quad (1)$$

Using CoPS, the label number distribution per antibody could be generated (Supplementary Fig. 3a,b), yielding a pseudo-ensemble DOL of labeled antibodies of 1.38, with 80% of labeled antibodies carrying a single fluorophore, 9% carrying two and the remaining antibodies carrying >2 fluorophores. Using the measured distribution of fluorophores/antibody, the fraction of unlabeled antibodies could be determined to be 8%.

## Data analysis
**ProDOL analysis.** *Segmentation.* Binary masks to segment cells from background were generated from the reference channel as input with a custom-written imageJ script (processAverageIJwiththunderSTORM.ijm). In brief, background was subtracted, high and low frequency domains were filtered from the image and a threshold was automatically generated. Finally, masks were exported as .tiff files for subsequent analysis.

*Emitter localization.* Single-molecule signals were localized with subpixel accuracy using ThunderSTORM called from a custom-written ImageJ script (processAverageIJwiththunderSTORM.ijm) with multi-emitter fitting enabled and a maximum number of three overlapping emitters. Localizations with sigma values outside a ±50% window around the modal PSF width were removed.

*Registration.* Registration of localizations in both target and reference channels were calculated in MATLAB (script: ProDOL_pipeline_thunderSTORM.m). One global transformation matrix for all images from the same condition was generated by registration of each localization set. Localization sets with <50 emitters were excluded. Additionally, all registrations with >3 pixels *xy* shift, >5° rotational shift and 5% scaling shift were excluded. All remaining transformation parameters were averaged and applied subsequently to all images in a given experiment.

*Threshold determination.* A distance cutoff *T* was calculated following equations (2)–(4) in MATLAB (script: ProDOL_pipeline_thunderSTORM.m) for all image sets contained in an experiment. Colocalization distance cutoff *T* at which the fraction of specific colocalization $F_c$ is maximized while the contribution of random colocalization $F_r$ is kept at a minimum.

$$F_c(t) = \frac{\sum_{\Delta d=0}^{t} N_c(\Delta d)}{N_{c,\text{total}}} \quad (2)$$

$$F_r(t) = \frac{\sum_{\Delta d=0}^{t} N_r(\Delta d)}{N_{r,\text{total}}} \quad (3)$$

$$z(T) = \max\{F_c(t) - F_r(t)\} \quad (4)$$

For each tolerance *t*, a specific colocalization score *z* was determined at which the fraction of colocalizing signals $N_c$ is compared with a randomized colocalization $N_r$. Random colocalization values are computed by rotating the target channel image by 90° relative to the reference channel image. $\Delta d$ represents the distance of reference and target signal.

*Density correction.* A density correction was performed using equation (5) to correct for missed localizations (script: ProDOL_pipeline_thunderSTORM.m). The fraction of missed emitter localizations and consequently the recovered DOL exhibited a linear dependency on the emitter density (Extended Data Fig. 2). Correction factors ($\text{CF}_{\text{slope}}$ and $\text{CF}_{\text{offset}}$) were determined from simulated data generated with testSTORM[41]. Simulated data were processed using the same approach as for experimental data described above and the recall as

function of simulated emitter density was assessed to obtain parameters for $CF_{slope}$ and $CF_{offset}$ using linear regression.

$$DOL = \frac{\text{Degree of colocalization}}{CF_{slope} \times \text{Density}_{measured} + CF_{offset}} \quad (5)$$

*Classification.* Manual classification of segmented data can be performed to remove image sets for which segmentation failed (script: imageSetInspector.m). Here segmented mask and reference channel are displayed in a random order to visually inspect segmentation results.

*Simulations for labeling efficiency measurements.* Randomly distributed single emitters were simulated with testSTORM. A Gaussian PSF model with 'Vesicles pattern' and acquisition parameters in line with experimental data was generated with drift disabled. 'Alexa Fluor 647' dye model was adapted with the following parameters to resemble emitter properties matching experimental data: on time: 0.025; off time: 0.01; bleaching constant: 0.2; emitted photon $s^{-1}$: 350; number of labels per epitope: 1. Background was selected from a compilation of representative images of unlabeled cells imaged under identical acquisition settings as used for ProDOL. Low-background images were acquired under TIRF illumination at 561 nm. High background images were acquired under TIRF at 640 nm excitation.

*Emitter number estimation with CoPS.* A Python library (pycops) was used to convert the raw.ptu data to .MBin files that contain multiple photon detections per laser pulse. The data from the first five to ten million laser pulses were then fitted to a mathematical model via a Levenberg–Marquardt algorithm. The data were collected and filtered for failed fits with a photon detection probability <0.00075 or <0.002, for SNAP-AF647 and SNAP-SiR, respectively. For cellular DOL determination of Nup107-SNAP, the median number of detected emitters was divided by 32 (known stoichiometry).

*Emitter number estimation with quickPBSA.* A custom Fiji script with the Thunderstorm plugin was used to determine the location of molecular clusters, after which the quickPBSA Python library was used to extract the intensity traces and correct for background intensity around each individual spot. The script also fitted the bleaching steps before summing up all the results. PBSA analysis results were filtered based on quality parameters output by the trace extraction algorithm. For cellular DOL determination of Nup107-SNAP, the median number of detected emitters was divided by 32 (known stoichiometry). The code is available at https://github.com/JohnDieSchere/quickpbsa.

*Labeling efficiency measurements with ELE.* All dSTORM data were analyzed in SMAP[42] using the fit_fastsimple workflow (dynamic factor: 1.3, with sCMOS correction and asymmetry enabled). The resulting localizations were analyzed using the 'SMAP_manual_NPC' manual related to ref. 10. Localization were filtered (locprec: 0–12 nm, PSF: 110–185 nm, frame: 500–Inf; and asymmetry: 0–0.2) and a NPC radius of 55 nm was assumed. NPC fitting was performed for all NPC with ≥3 labeled corners and a minimum localiation count of ≥2 emission events per corner.

## Statistics
**Curve fitting.** All quantitative SLP-76 and phospho-SLP-76 distributions determined using CoPS (Fig. 4) were fitted using a log-normal distribution in GraphPad Prism v9.5.0 (730). SLP-76 does not form a cluster of consistent copy number and processes that involve a multiplicative product of independent random variables lead to a log-normal distribution ($R^2 > 0.85$ for all fits).

Linear regression of detected fluorophores over DOLs was calculated in GraphPad Prism using 'simple linear regression' using mean, standard deviation and $N$ for fluorophore numbers and mean and standard error for DOL measurements.

**Statistical analysis.** Comparison between DOL determination methods (Fig. 2) was performed using nonparametric analysis of variance (Kruskal–Wallis). In time series measurements (Fig. 4a), significance test of linear fits was determined using analysis of covariance comparing the slope of linear fits to each other. phospho-SLP-76 histograms in the time series experiments were analyzed using nonparametric analysis of variance (Kruskal–Wallis). Levels of SLP-76 and phospho-SLP-76 in Nef experiments were compared with controls using nonparametric Kolmogorov–Smirnov. Significance testing of the phospho-SLP-76 to SLP-76-HaloTag ratio (Fig. 4) was determined using a two-sample $z$-test.

**Plotting.** Three-dimensional plots were generated in RStudio using the plotly library. Box plots span the interval from the 25th to the 75th percentile, with the median indicated by a horizontal line within the box. Whiskers extend to 1.5× the interquartile range.

### Reporting summary
Further information on research design is available in the Nature Portfolio Reporting Summary linked to this article.

### Data availability
The data that support the findings of this study are available from the corresponding author upon reasonable request.

### Code availability
The ProDOL analysis package was written in the MATLAB and tested under version: 9.11.0 (R2021b). The ProDOL software is freely available under GNU General Public License v3.0 (https://github.com/hertenlab/ProDOL). Additional acquisition and analysis routines are available from the corresponding author upon reasonable request.

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

### Acknowledgements
We acknowledge funding from the Deutsche Forschungsgemeinschaft through project PhotoQuant HE4559/6-1, by the Centre of Membrane Proteins and Receptors (Universities of Birmingham and Nottingham), and by the Academy of Medical Sciences (grant APR2\1013). O.T.F. and D.-P.H. acknowledge shared funding by the Bundesministerium für Bildung und Forschung (BMBF/VDI, Switch-Click). D.-P.H. also acknowledges funding by the Bundesministerium für Bildung und Forschung (BMBF/VDI, LungSys). U.K. acknowledges funding by the Deutsche Forschungsgemeinschaft within SFB/TRR186/2-A24 and

by the German Ministry for Education (BMBF) within the German Center for Lung Research (DZL) (82DZL004C4) and the LiSyM-Cancer networks SMART-NAFLD (031L0256A) and C-TIP-HCC [031L0257C]. Some computations described in this paper were performed using the University of Birmingham's BEAR Cloud service. We are grateful for the services provided by the FACS sorting facility at ZMBH (RI_00566) at Heidelberg University. We thank R. Bartenschlager for experimental support with the Huh7.5 cell line. U2-OS cells stably expressing Nup107-SNAP-tag were kindly provided by J. Ellenberg (EMBL Heidelberg).

## Author contributions

J.E., S.A.T. and K.Y. performed all data analyses. J.E., S.A.T., K.Y., S.H., F.H., W.C., S.G. and Z.S. acquired the data. S.H. generated the ProDOL probe. S.H., F.H., K.Y., J.E., J.H. and S.A.T. generated the ProDOL analysis workflow. W.C., S.H., N.T. and F.S. generated stable cell lines. Conceptualization, supervision, resources and project administration on T cell work were performed by O.T.F. and D.-P.H. O.T.F. and U.K. contributed general ideas and concepts. S.H. and D.-P.H. conceived the method. J.E., S.A.T., K.Y., O.T.F. and D.-P.H. wrote the paper with input from all authors.

## Competing interests

The authors declare no competing interests.

## Additional information

**Extended data** is available for this paper at https://doi.org/10.1038/s41592-024-02376-6.

**Correspondence and requests for materials** should be addressed to Dirk-Peter Herten.

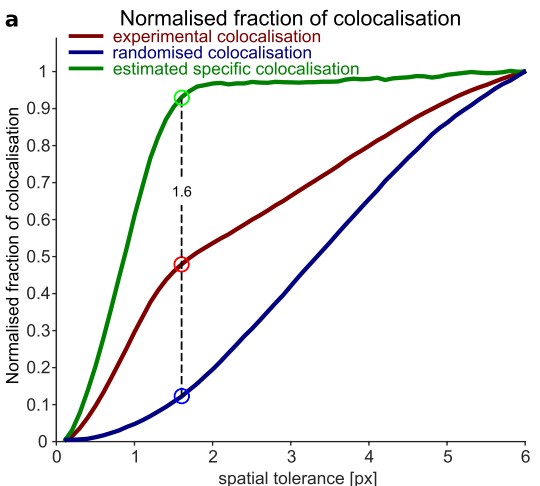

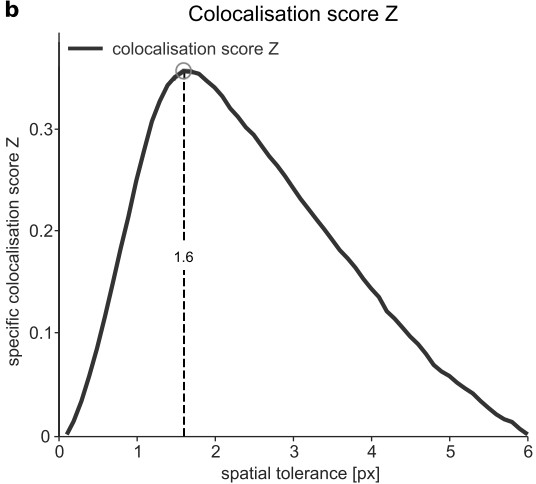

**Extended Data Fig. 1 | Representative plot for colocalisation threshold determination.** Vertical line represents the cut-off distance T at which a maximal specific colocalization score Z is observed. **a**, Normalised fraction of the colocalisations found in the experimental data (red), the randomised data (blue), and specific colocalisation (green) estimated from the difference of the total number of colocalisations found in experimental and randomised data. **b**, colocalisation score Z for data from (a) using equation (4) (see Methods).

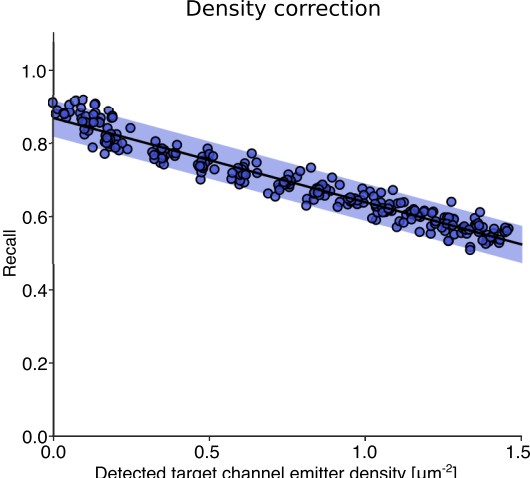

**Extended Data Fig. 2 | ProDOL density correction.** Density correction factors for raw DOL values determined from simulated data for emitter densities ranging from 0 to 1.5 µm$^{-2}$. Emitter images at defined densities were simulated and combined with images acquired from unlabelled samples to simulate background signal as described in the Methods section. 20 images per simulated point density were processed with the ProDOL data analysis pipeline and detected emitter localizations were compared against the truth emitter positions to compute the recall for each image (circles). Linear fits (black lines) were performed to obtain the density correction factor CF$_{slope}$ and offset value CF$_{offset}$. Shaded regions represent 95% confidence interval from linear fit.

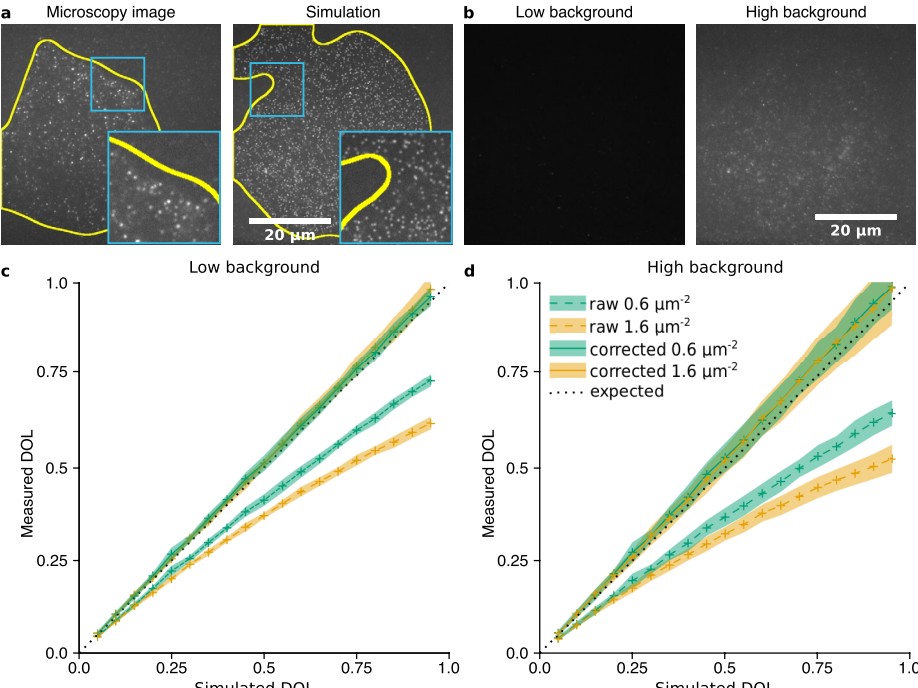

**Extended Data Fig. 3 | Validation of ProDOL labelling efficiency measurements using simulated data. a**, Comparison of experimental (left) and simulated (right) microscopic data. Simulated images were generated by combining experimental data of cells expressing ProDOL without tag labelling and simulated emitters closely resembling experimental data at varying density and background. Yellow outline represents a segmented cell. **b**, Example of high and low background from unlabelled H838 used in simulations. **c, d**, Simulated images were used to validate density corrections at reference density of 0.6 and 1.6 μm⁻² respectively (representative average emitter densities in cells). Median ± standard deviation for raw data (dashed) and corrected (solid) DOL. n = 20 simulated images per datapoint.

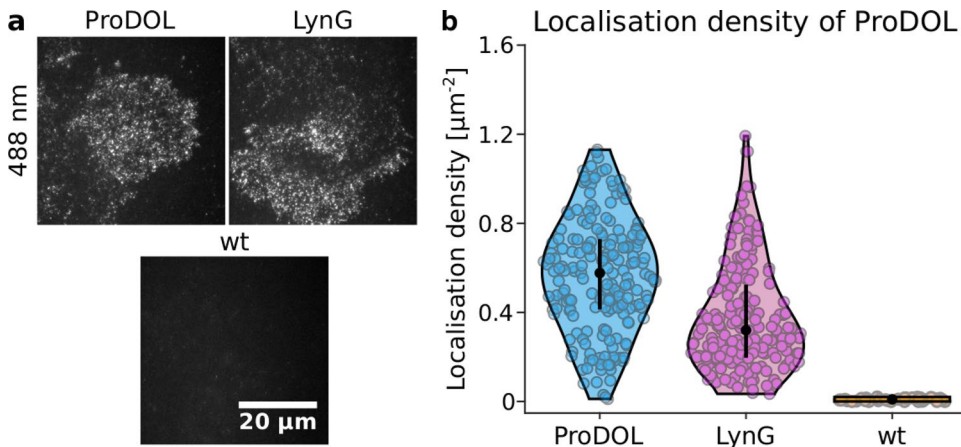

**Extended Data Fig. 4 | Comparison of detected emitter densities in the reference channel after localisation filtering.** Huh7.5-ProDOL, Huh7.5-LynG and Huh7.5 wild-type (wt) cells were prepared and imaged under identical ProDOL conditions. **a**, Representative images from each cell type, displayed using identical settings. **b**, Detected emitter densities shown as total number of detected localisation per cells divided by the segmented area. Circles represent individual cells. Data from 194 (ProDOL), 189 (LynG) and 38 (wt) cells.

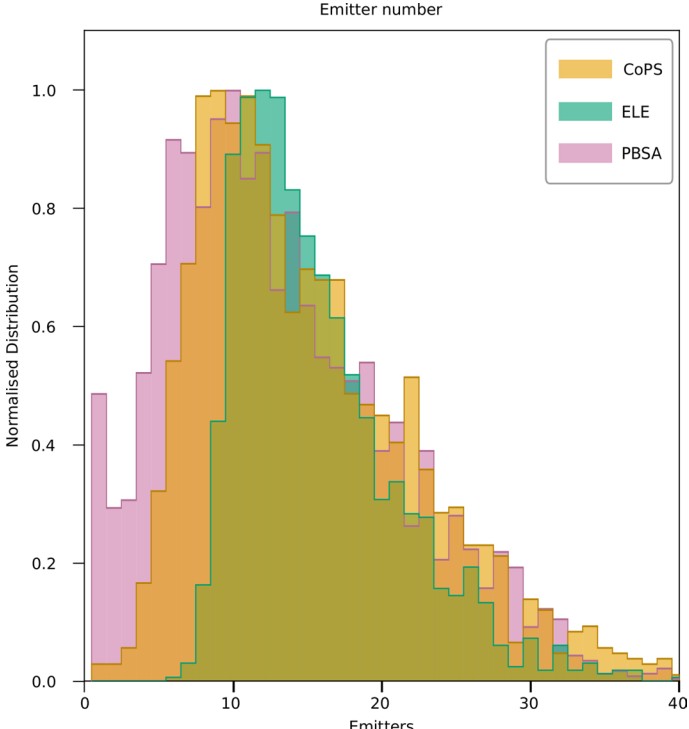

**Extended Data Fig. 5 | Label number distribution of Nup107-SNAP per NPC as measured by CoPS, ELE and PBSA.** Data represents distribution of labels per individual nuclear pore complex (NPC) as determined by CoPS (orange), ELE (green), or PBSA (purple). ELE has a low detection rate at <8 emitters as a circular fit needs to be performed and therefore needs multiple corners labelled. ELE and PBSA both have detections of >32 labels, indicating counting of multiple NPC not detected in diffraction limited microscopy. n=2352, 3668, and 1626 NPCs for the ELE, quickPBSA, and CoPS, respectively.

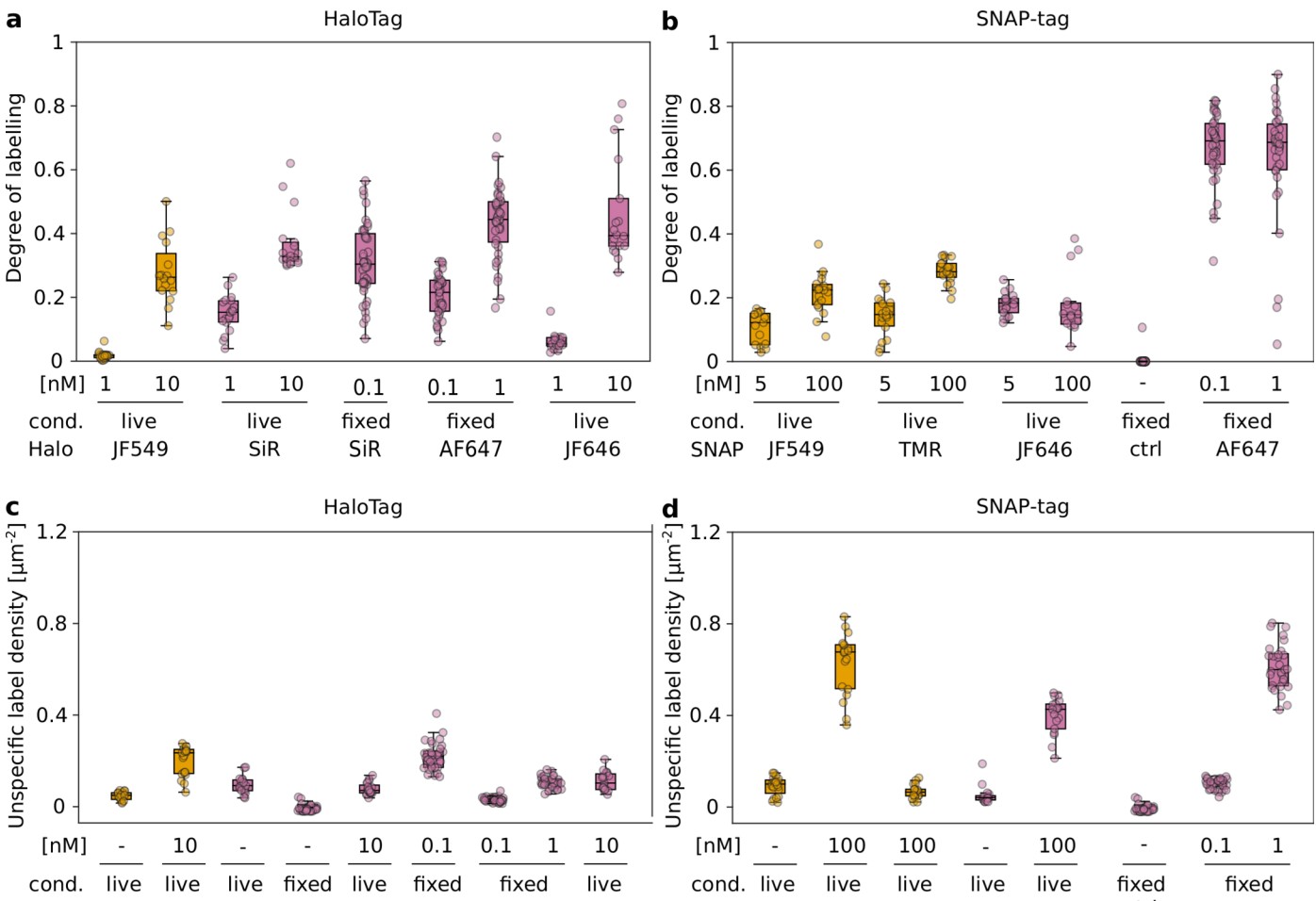

**Extended Data Fig. 6 | Labelling efficiency and unspecific label density in Huh-7.5 cells.** Huh-7.5 cells expressing ProDOL for DOL determination (**a**,**b**) or LynG for unspecific label density determination (c,d) labelled either while live or fixed. Cells were labelled for both HaloTag (**a**,**c**) and SNAP-tag (**b**,**d**) with varying dyes (live: 30 min, fixed: 120 min). Data points represent DOL or label density per cell, 15–42 cells per condition. Orange: 561 nm excitation, purple: 640 nm excitation. Box plots span the interval from the 25th to the 75th percentile with the median indicated by a horizontal line within the box. Whiskers extend to 1.5× the interquartile range.

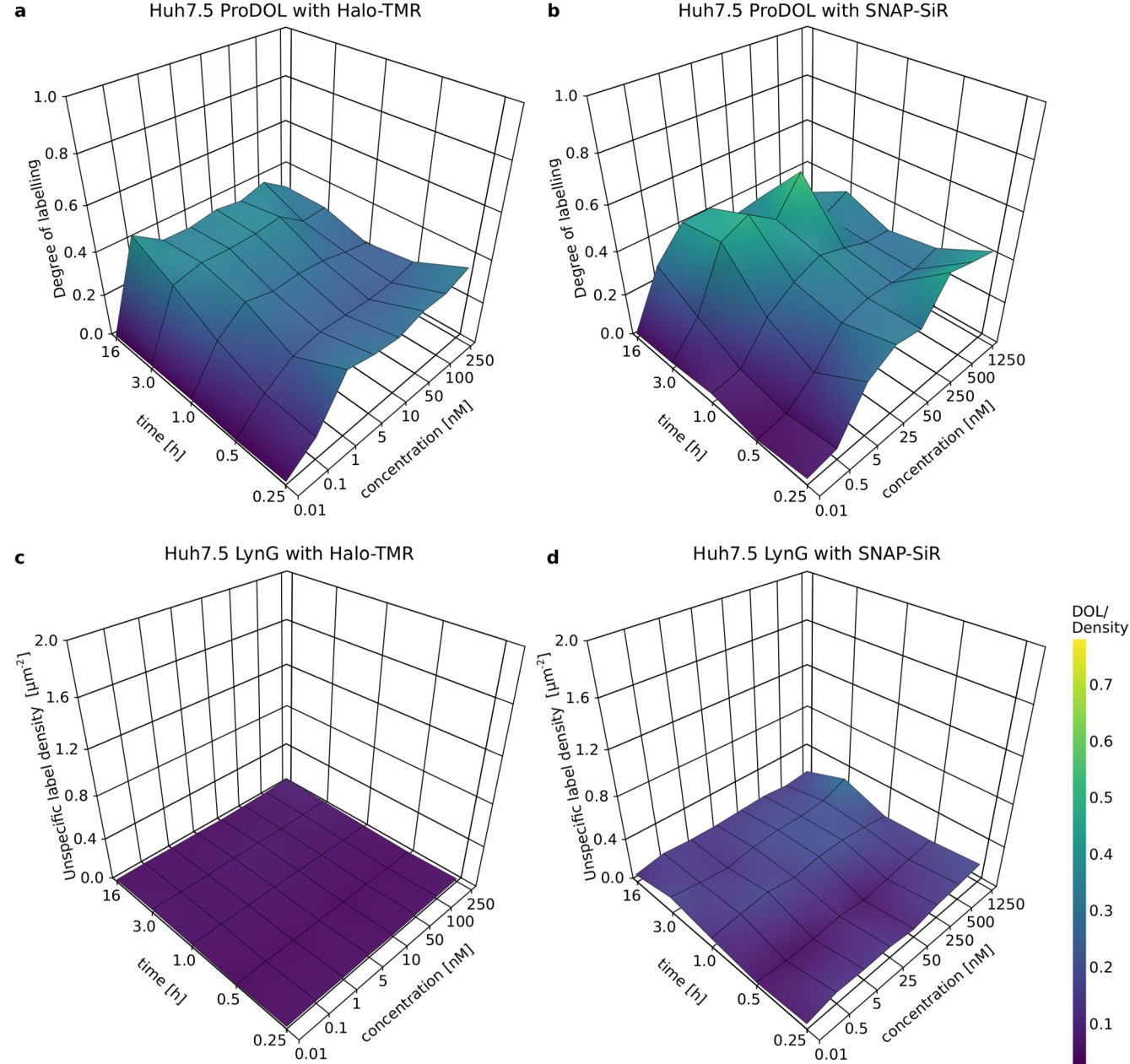

**Extended Data Fig. 7 | Determination of DOL as a parameter of incubation time and dye concentration. a**, The same samples of Huh7.5 cells expressing ProDOL (a, b) and LynG (c, d) were stained with varying concentrations of Halo-TMR (a,c) and SNAP-SiR (b,d). **a,b,** Determined median DOL using ProDOL at varying ligand concentrations and incubation time. **c,d**, Unspecific labelling density measured at varying ligand concentrations and incubation time.

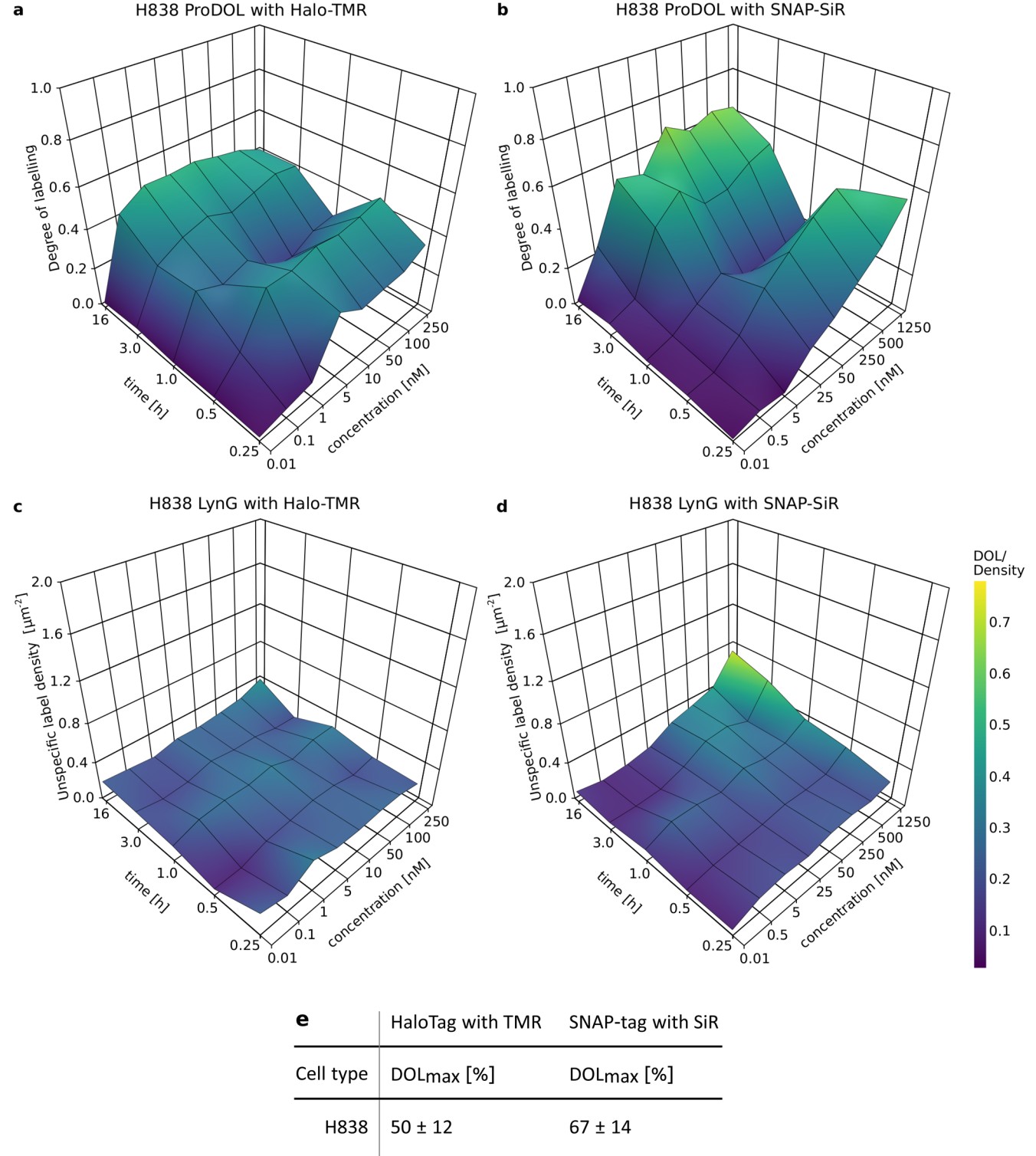

**Extended Data Fig. 8 | Determination of DOL as a parameter of incubation time and dye concentration. a**, The same samples of H838 cells expressing ProDOL (a, b) and LynG (c, d) were stained with varying concentrations of Halo-TMR (a,c) and SNAP-SiR (b,d). **a,b**, Determined median DOL using ProDOL at varying ligand concentrations and incubation time. **c,d**, Unspecific labelling density measured at varying ligand concentrations and incubation time.

**e**, Maximum DOL found for Halo-TMR and SNAP-SiR per cell line. Median from 4–15 cells per condition. Note that DOL measurements for all concentration at 1 h incubation time appear to have a lower labelling efficiency due to experimental variations. This highlights the importance of ProDOL as an easy to implement control experiment to validate consistent labelling.

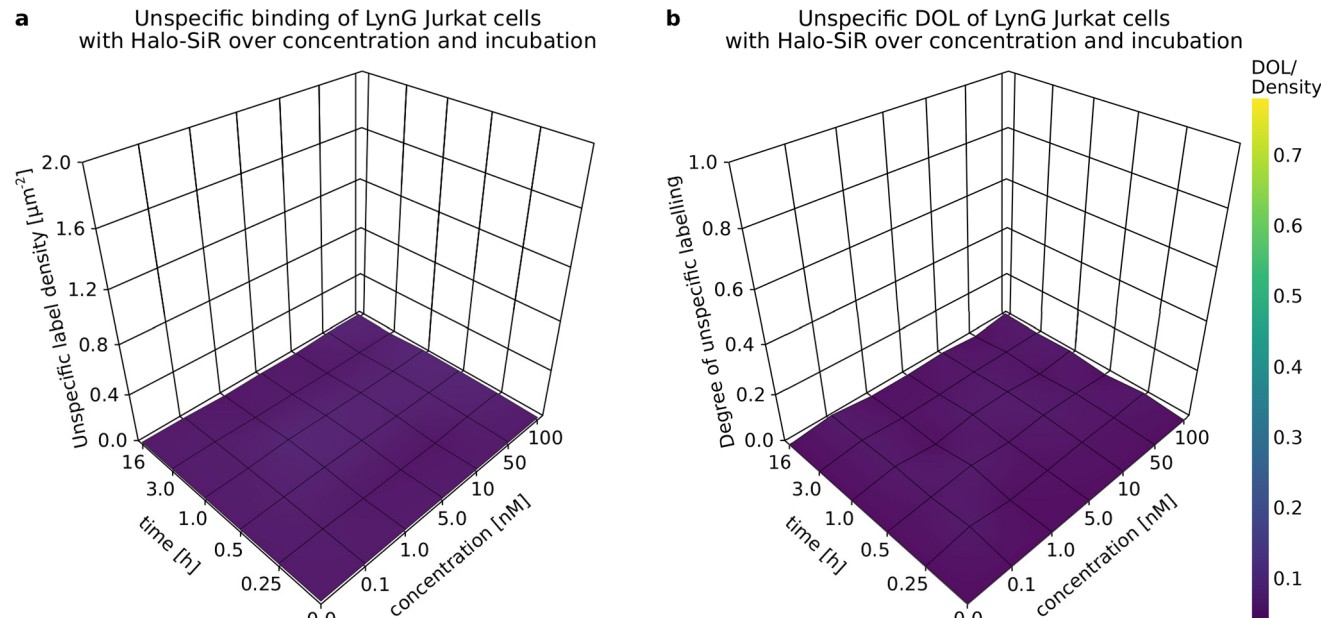

**Extended Data Fig. 9 | Unspecific binding and unspecific DOL of Halo-SiR in Jurkat cells expressing the LynG construct. a**, Unspecific labelling density and **b**, apparent DOL due to unspecific labels measured at varying ligand concentrations and incubation time.

# Reporting Summary

## Statistics

For all statistical analyses, confirm that the following items are present in the figure legend, table legend, main text, or Methods section.

| n/a | Confirmed | |
|---|---|---|
| ☐ | ☒ | The exact sample size (*n*) for each experimental group/condition, given as a discrete number and unit of measurement |
| ☐ | ☒ | A statement on whether measurements were taken from distinct samples or whether the same sample was measured repeatedly |
| ☐ | ☒ | The statistical test(s) used AND whether they are one- or two-sided *Only common tests should be described solely by name; describe more complex techniques in the Methods section.* |
| ☐ | ☒ | A description of all covariates tested |
| ☐ | ☒ | A description of any assumptions or corrections, such as tests of normality and adjustment for multiple comparisons |
| ☐ | ☒ | A full description of the statistical parameters including central tendency (e.g. means) or other basic estimates (e.g. regression coefficient) AND variation (e.g. standard deviation) or associated estimates of uncertainty (e.g. confidence intervals) |
| ☐ | ☒ | For null hypothesis testing, the test statistic (e.g. *F*, *t*, *r*) with confidence intervals, effect sizes, degrees of freedom and *P* value noted *Give P values as exact values whenever suitable.* |
| ☒ | ☐ | For Bayesian analysis, information on the choice of priors and Markov chain Monte Carlo settings |
| ☒ | ☐ | For hierarchical and complex designs, identification of the appropriate level for tests and full reporting of outcomes |
| ☒ | ☐ | Estimates of effect sizes (e.g. Cohen's *d*, Pearson's *r*), indicating how they were calculated |

*Our web collection on statistics for biologists contains articles on many of the points above.*

## Software and code

Policy information about availability of computer code

| Data collection | MicroManager 1.4.22<br>MicroManager 2.0.1<br>SymPhoTime 64 2.7 |
|---|---|
| Data analysis | ProDOL analysis pipeline (https://github.com/hertenlab/ProDOL)<br>GraphPad Prism v9.5.0 (730)<br>MATLAB v9.11.0<br>Rstudio v1.3.959 with R version 4.0.2<br>SMAP (https://github.com/jries/SMAP)<br>Fiji (https://imagej.net/software/fiji/downloads) version 1.54<br>thunderSTORM (https://zitmen.github.io/thunderstorm/) version 1.3<br>quickPBSA (https://github.com/JohnDieSchere/quickpbsa) version 2021.0.1<br>TestSTORM (https://titan.physx.u-szeged.hu/~adoptim/?page_id=183) version 2.0<br>pycops (unreleased code) |

For manuscripts utilizing custom algorithms or software that are central to the research but not yet described in published literature, software must be made available to editors and reviewers. We strongly encourage code deposition in a community repository (e.g. GitHub). See the Nature Portfolio guidelines for submitting code & software for further information.

## Data

Policy information about availability of data

All manuscripts must include a data availability statement. This statement should provide the following information, where applicable:

- Accession codes, unique identifiers, or web links for publicly available datasets
- A description of any restrictions on data availability
- For clinical datasets or third party data, please ensure that the statement adheres to our policy

The authors declare that the data supporting the findings of this study are available within the paper and its Supplementary Information files. Should any raw data files be needed in another format they are available from the corresponding author upon reasonable request.

## Human research participants

Policy information about studies involving human research participants and Sex and Gender in Research.

| | |
|---|---|
| Reporting on sex and gender | N/A |
| Population characteristics | N/A |
| Recruitment | N/A |
| Ethics oversight | N/A |

Note that full information on the approval of the study protocol must also be provided in the manuscript.

# Field-specific reporting

Please select the one below that is the best fit for your research. If you are not sure, read the appropriate sections before making your selection.

☒ Life sciences ☐ Behavioural & social sciences ☐ Ecological, evolutionary & environmental sciences

For a reference copy of the document with all sections, see nature.com/documents/nr-reporting-summary-flat.pdf

# Life sciences study design

All studies must disclose on these points even when the disclosure is negative.

| | |
|---|---|
| Sample size | No statistical methods to determine sample sizes were used. Unless stated otherwise, sample number refers to the number of cells analyzed per condition. |
| Data exclusions | Data was acquired by automated microscopy and cells lacking specific eGFP expression were excluded after acquisition and before DOL analysis. Additionally, the ProDOL software includes a visual inspection software to remove cells where segmentation was not successful. Other data was not excluded unless stated otherwise in the text. |
| Replication | Data was obtained from multiple experiments as detailed in the manuscript and the supplemental information. Reported results could be replicated across multiple experiments with replicates generating similar results. |
| Randomization | No randomization was used in the experiments. |
| Blinding | Blinding was not possible due to the nature of the study. |

# Reporting for specific materials, systems and methods

We require information from authors about some types of materials, experimental systems and methods used in many studies. Here, indicate whether each material, system or method listed is relevant to your study. If you are not sure if a list item applies to your research, read the appropriate section before selecting a response.

## Materials & experimental systems

| n/a | Involved in the study |
|---|---|
| ☐ | ☒ Antibodies |
| ☐ | ☒ Eukaryotic cell lines |
| ☒ | ☐ Palaeontology and archaeology |
| ☒ | ☐ Animals and other organisms |
| ☒ | ☐ Clinical data |
| ☒ | ☐ Dual use research of concern |

## Methods

| n/a | Involved in the study |
|---|---|
| ☒ | ☐ ChIP-seq |
| ☒ | ☐ Flow cytometry |
| ☒ | ☐ MRI-based neuroimaging |

# Antibodies

| Antibodies used | 1. α-SLP76-PY145 (Rabbit) - monoclonal IF - Abcam (ab206782) - clone EP2853Y - 2 µg/ml<br>2. α-GFP (Mouse) - polyclonal WB - Cell Biolabs (part 212101) - n/a -1:1000 dilution<br>3. α-CD3 (Mouse) - monoclonal surface preparation - BioLegend (V CD03.05) -clone HIT3a - 1:50 dilution at 10 µg/ml<br>4. α-mouse-IgG conjugated with HRP (Goat) - polyclonal WB - Jackson Immuno- Research (115-035-003) - n/a - 1:10000 dilution |
|---|---|
| Validation | All antibodies were commercially available and validated by the manufacturer. Furthermore, they have been used in previous publication and staining with all antibodies yielded the expected spatial distribution/localisation of the protein of interest.<br>1. The antibody has been validated for FlowCyt, WB, Dot blot and IF by the manufacturer.<br>2. Validated by ELISA by the manufacturer.<br>3.FC -Quality tested. Activ -Reported in the literature:<br> Sedelies KA, et al. 2004. J. Biol. Chem. 279:26581. (Activ)<br> Rivollier A, et al. 2004. Blood 104:4029. (Activ)<br> Scharschmidt E, et al. 2004. Mol. Cell Biol. 24:3860. (Activ)<br> Smeltz RB. 2007. J. Immunol. 178:4786. (Activ)<br>4. Based on immunoelectrophoresis and/or ELISA, the antibody reacts with whole molecule mouse IgG. It also reacts with the light chains of other mouse immunoglobulins. No antibody was detected against non-immunoglobulin serum proteins. The antibody may cross-react with immunoglobulins from other species as reported by the manufacturer. |

# Eukaryotic cell lines

Policy information about cell lines and Sex and Gender in Research

| Cell line source(s) | 1. H838 (NCI-H838, ATCC)<br>2. HeLa (CCL-2,ATCC)<br>3. HEK293T were obtained from the laboratory of Ralf Bartenschlager (Heidelberg University)<br>4. Jurkat cells were obtained from the laboratory of Oliver Fackler (University Hospital Heidelberg)<br>5. U2OS cells were obtained from the laboratory of Jan Ellenberg (EMBL Heidelberg).<br>6. Huh-7.5 cells were obtained from the laboratory of Ralf Bartenschlager (Heidelberg University) |
|---|---|
| Authentication | 1. Cell line authentication was performed using Multiplex Cell Authentication by Multiplexion (Heidelberg, Germany) as described (Castro et al, 2013).<br>2. Cell line authentication was performed using Multiplex Cell Authentication by Multiplexion (Heidelberg, Germany) as described (Castro et al, 2013).<br>3. Cell line was not authenticated<br>4. Cell line was not authenticated<br>5. Authenticated via STR profiling by manufacturer<br>6. Cell line authentication was performed using Multiplex Cell Authentication by Multiplexion (Heidelberg, Germany) as described (Castro et al, 2013).<br>Castro F, Dirks WG, Fahnrich S, Hotz-Wagenblatt A, Pawlita M, Schmitt M (2013) High-throughput SNP-based authentication of human cell lines. Int J Cancer 132: 308 − 314 |
| Mycoplasma contamination | Cell lines have been regularly tested for mycoplasma contamination with negative results. |
| Commonly misidentified lines<br>(See ICLAC register) | No commonly misidentified cell lines were used in the study. |

