## [Peer Review File · Nature Methods]

Peer Review Information

Manuscript Title: ProDOL: A general method to determine the degree of labelling for staining optimisation and molecular counting.

Corresponding author name(s): Dirk-Peter Herten

Editorial Notes: None

Reviewer Comments & Decisions:

Decision Letter, initial version:

Dear Dirk-Peter,

Your Article, "A General Method to Accurately Count Molecular Complexes and Determine the Degree of Labelling in Cells Using Protein Tags", has now been seen by two reviewers. As you will see from their comments below, although the reviewers find your work of considerable potential interest, they have raised a number of concerns. We are interested in the possibility of publishing your paper in Nature Methods, but would like to consider your response to these concerns before we reach a final decision on publication. We therefore invite you to revise your manuscript to address these concerns.

We found the comments on the whole constructive and reasonable. When revising, we ask that you experimentally validate the assumption about DOL of the probe for PM labeling matching intracellular labeling. We ask that you discuss applicability beyond Halo/SNAP tags. We also ask that you clarify the underlying math and make sure the methods description is sufficient for users to easily reproduce it.

* include a point-by-point response to the reviewers and to any editorial suggestions

* please underline/highlight any additions to the text or areas with other significant changes to facilitate review of the revised manuscript

* address the points listed described below to conform to our open science requirements

* ensure it complies with our general format requirements as set out in our guide to authors at www.nature.com/naturemethods

* resubmit all the necessary files electronically by using the link below to access your home page

[Redacted]

We hope to receive your revised paper within three months. If you cannot send it within this time, please let us know. In this event, we will still be happy to reconsider your paper at a later date so long as nothing similar has been accepted for publication at Nature Methods or published elsewhere.

OPEN SCIENCE REQUIREMENTS

REPORTING SUMMARY AND EDITORIAL POLICY CHECKLISTS

Please note that these forms are dynamic ‘smart pdfs’ and must therefore be downloaded and completed in Adobe Reader. We will then flatten them for ease of use by the reviewers. If you would like to reference the guidance text as you complete the template, please access these flattened versions at <http://www.nature.com/authors/policies/availability.html>.

IMAGE INTEGRITY

When submitting the revised version of your manuscript, please pay close attention to our Digital Image Integrity Guidelines and to the following points below:

DATA AVAILABILITY

All novel DNA and RNA sequencing data, protein sequences, genetic polymorphisms, linked genotype and phenotype data, gene expression data, macromolecular structures, and proteomics data must be deposited in a publicly accessible database, and accession codes and associated hyperlinks must be provided in the “Data Availability” section.

Please include a “Data availability” subsection in the Online Methods. This section should inform readers about the availability of the data used to support the conclusions of your study, including accession codes to public repositories, references to source data that may be published alongside the paper, unique identifiers such as URLs to data repository entries, or data set DOIs, and any other statement about data availability. At a minimum, you should include the following statement: “The data that support the findings of this study are available from the corresponding author upon request”, describing which data is available upon request and mentioning any restrictions on availability. If DOIs are provided, please include these in the Reference list (authors, title, publisher (repository name), identifier, year). For more guidance on how to write this section please see: <http://www.nature.com/authors/policies/data/data-availability-statements-data-citations.pdf>

CODE AVAILABILITY

Please include a “Code Availability” subsection in the Online Methods which details how your custom code is made available. Only in rare cases (where code is not central to the main conclusions of the paper) is the statement “available upon request” allowed (and reasons should be specified).

MATERIALS AVAILABILITY

SUPPLEMENTARY PROTOCOL

To help facilitate reproducibility and uptake of your method, we ask you to prepare a step-by-step Supplementary Protocol for the method described in this paper. We encourage authors to share their step-by-step experimental protocols on a protocol sharing platform of their choice and report the protocol DOI in the reference list. Nature Portfolio 's Protocol Exchange is a free-to-use and open resource for protocols; protocols deposited in Protocol Exchange are citable and can be linked from the published article. More details can found at www.nature.com/protocolexchange/about.

ORCID

Sincerely,
Rita

Rita Strack, Ph.D.
Senior Editor

Nature Methods

Reviewers' Comments:

Reviewer #1:

Remarks to the Author:

Review of "A General Method to Accurately Count Molecular Complexes and Determine the Degree of Labelling in Cells Using Protein Tags" by Tashev et al.

Introduction

In their manuscript, Tashev et al. introduce a simple and direct method for measuring the labeling efficiency of self-labeling enzyme tags such as SNAP and HALO, which are widely used in fluorescence microscopy. The concept, ProDOL, is based on a protein construct composed of a membrane-bound GFP and two tags: SNAP and HALO. Using the GFP fluorescence as a reference marker, single molecule imaging of cells expressing this probe allows a direct measurement of the absolute fraction of a given target tag which is labeled with a fluorophore in the prepared sample. The authors use this approach to optimize the SNAP and HALO labeling protocols, quantitatively examining the effects of parameters such as substrate concentrations, incubation times, and fixation. Next, knowing the expected SNAP and HALO labeling efficiencies, they were able to quantify the time-dependent copy numbers of two different proteins (SLP76 and Nef) involved in T-cell receptor signaling, under various conditions.

General comments

The paper is very well written and the new method is simple and powerful. Once establishing the approach, the authors were immediately able to optimize the SNAP and HALO labeling protocols in a manner which was not possible before, and this alone is a major advance, but it also speaks to the utility of the new method. Importantly, the authors have benchmarked their results against several other earlier approaches, obtaining consistent results.

A minor criticism I would put forward is this. The new methodology of the paper is clearly the establishment of the DOL measurement protocol. Yet, this is not emphasized in the title of the paper, which rather suggests a new method for accurate counting of proteins as the main point. Also, at the end of the manuscript, the context and implications of the highly detailed study of SLP76 and Nef copy numbers in TCR signaling become lost, due to limitations of space, and this section is perhaps trying to squeeze too much into the story, detracting from the overall message.

Specific comments

1. The ProDOL probe incorporates GFP as a reference, and two tags: SNAP and HALO. However, it seems that the same approach could be used to measure the labeling efficiency for other probes, such as nanobodies, antibodies, etc. How can this method be adapted, or generalized, so that it may be used to measure DOL for contexts other than SNAP or HALO? Can this question be addressed in the manuscript?
2. In Figure 2, the authors show a quantitative validation of the method, benchmarked versus three other techniques. The data appear much more spread for ProDOL, when compared with the others. Why is this? The ProDOL concept seems to be simple and straightforward, so it is difficult to see where the source of error is entering into the DOL calculation. Can the authors comment on, and/or improve this aspect? Is this a case of the density of the ProDOL expression being too high for a precise measurement? In the plot, each point represents the result for a single cell, suggesting there is a large cell to cell variability. Is there also variability for the data within each cell? E.g. by taking sub-datasets from within a cell, could this be tested? The aim of this remark is, again, to determine the source of the larger error bar.
3. Regarding the optimization of the labeling conditions, it appears in Figure 3b that the best DOL was obtained for 16 hours incubation time. How was non-specific labeling affected by incubation time? Based on their data, can the authors present a single set of labeling conditions (concentration, time, etc) which yield the highest DOL with the lowest non-specific background for each probe?
4. In Supplementary Figure 4 panels a and b, there is a pronounced dip in the DOL vs incubation time plot at a time of 1 hour. Does this make sense?
5. How accurate is the threshold determination / density correction procedure presented in the methods section? Can the authors validate this procedure with experimental data (e.g. with cells expressing very low densities of ProDOL)? Do errors introduced at this step contribute significantly to the variance of the DOL result?
6. The new approach to measuring DOL is based on the assumption that the DOL of a protein of interest in its cellular context will be the same as the DOL of the ProDOL probe at the plasma membrane. The authors correctly questioned this assumption in the discussion section of the manuscript, and note that their benchmarking results for Nup107 would support this argument. Could the authors also test this by targeting the ProDOL probe to another region of the cell? Can the authors think of another benchmarking target (e.g. in the nuclear interior, at a mitochondrion, etc.) which could be used to verify the assumption, similar to their Nup107 data? This comment is not necessarily asking for any new data or experiments, but should be seen rather as a question for the authors to comment on, and possibly a suggestion to slightly expand this area of the discussion.

7. The application of the ProDOL method to TCR signaling, presented in the last section of the main text, is extensive and detailed. While I appreciate that this data fully demonstrates the validity of the new approach, adding too much detail here risks losing the reader and detracting from the main message of the paper, which for me seems to be the new method for measuring labelling efficiency. Also, the importance of the new biology which is presented here risks becoming lost because there is simply not sufficient space to do it justice. Is there a way to simplify this section of the manuscript, and Figure 4, while maintaining the demonstration of the ProDOL method in a real biology application?

Conclusion

The manuscript presents an exciting new method, which the authors have applied to measure the absolute labeling efficiency of SNAP and HALO tags, and furthermore to optimize existing labeling protocols for these tags. The authors also demonstrate counting of absolute protein numbers present in protein microclusters during TCR signaling. I expect this study to be of significant benefit to the fluorescence imaging field, and I would recommend publication when the reviewers' comments are addressed.

Reviewer #2:

Remarks to the Author:

This manuscript by Tashev et al. presented a new way to evaluate the degree of labeling (DOL) for protein-tag (e.g. SNAP-tag, Halo-tag) labeling. The authors first fused SNAP-tag and Halo-Tag to a membrane anchored reference fluorescent protein (eGFP in this work). The DOL of the protein-tag labeling can then be determined by the percentage of reference fluorescent protein that are colocalized with the labelled protein-tags. Compared to other similar works using antibody or another protein-tag as references, current approach used fused fluorescent protein as reference. It could avoid the effect of unspecific labeling of the reference signal which could result in an underestimation of DOL. The authors showed different DOL of the SNAP-tag and Halo-tag for different labeling approaches (e.g. different dye concentrations, incubation time fixation conditions). Combined with CoPS, the authors also used this approach to measure the protein copy number in T cells. The precise determination of the DOL is very important for quantitative fluorescence microscopy. Although the current approach offers a potentially easy and versatile tool for this purpose, I am not convinced that it is generally applicable to different biological systems without the need for further validation.

Major Concern:

The DOL calibration probe is evaluated independently with the protein of interest (POI). Therefore, it has to assume that similar DOL of the protein-tag within the calibration probe and POI. Although the author showed similar DOL of the protein-tag with the plasma membrane anchored ProDOL probe and

nuclear pore complex, it is still not convinced for me that this method could be generally applicable to other systems. It is well known that many local environmental parameters could affect the maturation of the protein-tags and their accessibility to the fluorophores, such as intra- or extracellular location, fused proteins, and fixation conditions. Especially, the authors also observed significant variations of the DOL across different cell lines as show in Supplementary Fig. 4.

Other Comments:

1. In the main text line 241-244, the number of the phospho-SLP-76 was determined using CoPS. The author simply corrected the number with dye to antibody ratio. However, the binding efficiency of the antibody to phospho-SLP-76 was not known. It is therefore difficult to determine the copy number of phospho-SLP-76.
2. The effective labeling efficiency used in Ref. 10 is defined by the fraction of target proteins that carry a fluorophore that is detected as a usable localization. This parameter takes into account bleaching of the fluorophores or dim and nonfunctioning fluorophores during SMLM imaging. In theory, this value should be slightly lower than the DOL defined in this work.
3. The cut-off T to determine the colocalization is a bit too complex to me. Is it necessary to do so? Since the expression level of the ProDOL probe can be adjusted so that the distance between individual ProDOL probe is larger than the diffraction limit, it is actually quite easy to set the cut-off T less than the diffraction limit to determine whether reference signals and protein-tag signals are colocalized or not.
4. Since most experiments were performed on the plasma membrane near the coverslip, did the authors investigate the effect of the non-specific binding of the dyes on the coverslip?
5. Many parameters of the three equations in the methods were not defined. Equation 2 should be re-written.
6. Line 216-218, the authors mentioned that only cells that adhered to the cover slide within the first 3 seconds were observed, and all other cells washed away. I am curious how this can be performed. Can the authors give more information about this?

Author Rebuttal to Initial comments
--

Rebuttal

First of all, we thank the reviewers for their valuable feedback. Below you will find a point-by-point response to all comments.

1 Reviewer #1:

1.1 *The paper is very well written and the new method is simple and powerful. Once establishing the approach, the authors were immediately able to optimize the SNAP and HALO labeling protocols in a manner which was not possible before, and this alone is a major advance, but it also speaks to the utility of the new method. Importantly, the authors have benchmarked their results against several other earlier approaches, obtaining consistent results.*

Response 1.1: We appreciate the positive evaluation of our manuscript and are grateful for highlighting the potential of ProDOL for optimization of labelling protocols as well as for recognizing that benchmarking against alternative methods yielded highly consistent results.

1.2 *A minor criticism I would put forward is this. The new methodology of the paper is clearly the establishment of the DOL measurement protocol. Yet, this is not emphasized in the title of the paper, which rather suggests a new method for accurate counting of proteins as the main point.*

Response 1.2: We thank Reviewer #1 for raising this issue. We have changed the title of the revised manuscript to better reflect the content of our study which is indeed the establishment of ProDOL as tool for measuring labelling efficiencies. The updated title now reads: "ProDOL: A general method to determine the degree of labelling for staining optimisation and molecular counting."

1.3 *Also, at the end of the manuscript, the context and implications of the highly detailed study of SLP76 and Nef copy numbers in TCR signaling become lost, due to limitations of space, and this section is perhaps trying to squeeze too much into the story, detracting from the overall message.*

Response 1.3: We carefully studied our manuscript in light of this comment and ultimately decided to limit our focus to the assessment of SLP-76 copy numbers in signalling microclusters to showcase the usefulness of ProDOL in such experiments. As we feel that the complementary analysis of the phosphorylation state of SLP76 is of high interest to the T cell signalling community, we decided to retain these results in the manuscript, but to shift them to Supplementary Note 1 and two new Supplementary Figures (Supplementary Figs. 11, 12).

1.4 *The ProDOL probe incorporates GFP as a reference, and two tags: SNAP and HALO. However, it seems that the same approach could be used to measure the labeling efficiency for other probes, such as nanobodies, antibodies, etc. How can this method be adapted, or generalized, so that it may be used to measure DOL for contexts other than SNAP or HALO? Can this question be addressed in the manuscript?*

Response 1.4: We thank the Reviewer for this suggestion which motivated us to expand on this aspect of the discussion in greater detail. In addition to the possibility of using the ProDOL probe for characterising alternative protein tags as mentioned in the initial version of our manuscript, we have now included an extended discussion on the potential use of the ProDOL probe for characterising immunolabels such as nanobodies and antibodies. The updated discussion also includes a discussion of the limitations of the ProDOL concept when using affinity reagents as labels (lines 297-306 in the revised manuscript).

- 1.5 *In Figure 2, the authors show a quantitative validation of the method, benchmarked versus three other techniques. The data appear much more spread for ProDOL, when compared with the others. Why is this? The ProDOL concept seems to be simple and straightforward, so it is difficult to see where the source of error is entering into the DOL calculation. Can the authors comment on, and/or improve this aspect? Is this a case of the density of the ProDOL expression being too high for a precise measurement? In the plot, each point represents the result for a single cell, suggesting there is a large cell to cell variability. Is there also variability for the data within each cell? E.g. by taking sub-datasets from within a cell, could this be tested? The aim of this remark is, again, to determine the source of the larger error bar.*

Response 1.5: As described in the initial version of our manuscript, we observed a comparable single-cell DOL variability as typically measured by the standard deviation (SD) across cells for the different methods ($42.2 \pm 4.1\%$, $40.6 \pm 5.8\%$, $40.5 \pm 4.9\%$ and $42.6 \pm 5.3\%$ for ELE, PBSA, CoPS and ProDOL). The larger apparent spread of cell-wise DOL values for ProDOL reflected in the increased interquartile range of DOL estimates observed by the reviewer, prompted us to replot the cell-wise DOL values shown in Fig. 2f in a non-overlapping fashion to better display the full distribution of values for individual methods. We believe that the updated plots now make it easier to appreciate that the majority of cell-wise DOL values for ProDOL measurements indeed shows a spread comparable to that of the additional tested methods. This change also shows that the increased number of outliers observed is simply reflecting the larger sample size for ProDOL.

We thank the Reviewer for suggesting a sub-dataset analysis to access intra-cellular spread of the DOL in individual cells and to better understand potential sources of variation. To perform this analysis, we filtered the data presented in Fig. 2f of our manuscript to only include FOVs where each quadrant was covered by at least 20% of cellular surface. A subset of randomly chosen data was then used to run the ProDOL analysis pipeline on each of these quadrant images. We found that overall, the mean DOL recovered from the averaged quadrant analysis matches the result of an analysis run on full images (Fig. R1.5a). When computing the spread of recovered DOL values across all quadrants from all cells, we observe an increased variability, possibly due to limited numbers of localizations contained within individual quadrants (Fig. R1.5b). Here, the variability within quadrants from the same source image is smaller than the variability across all quadrants from all images (SDs of 4.5% vs. 6.0% respectively). Therefore, while there seems to be a measurable cell-to-cell variability aspect, it is not a dominating source of the uncertainty.

Regarding potential sources of cell-to-cell variations in measured DOLs, we agree with the reviewer, that minor variances between cells (e.g. in probe density, cell size or signal intensity) will lead to variation in the measured DOL per cell. To ensure that probe densities per cell fall into a permissible density regime covered by our validation using simulated data presented in Supplementary Fig. 4, we determined the probe densities per cell based on the GFP reference signal and added Supplementary Fig. 5 to the revised manuscript. This revealed densities of $0.75 \pm 0.33 \mu\text{m}^{-2}$ and $0.4 \pm 0.3 \mu\text{m}^{-2}$ for ProDOL probe and LynG probe confirming that the probe densities per cell fall into the validated regime and ensuring robust localization of individual emitters.

To account for cell-to-cell variations, we include at least 10 (typically ~20) cells per DOL calibration. In contrast to the other three tested methods (ELE, CoPS, PBSA), this does not pose a significant experimental hurdle for ProDOL as acquiring image stacks on individual cells typically takes less than one second while the data acquisition for the other methods will require ~100s of seconds per cell.

Figure R1.5: Sub-dataset analysis of ProDOL. Cells that had sufficient reference signal were split in quadrants, which were analysed independently. a, Comparison between cell-wide (red) and the average values of the four quadrants (blue). b, Boxplot showing the distribution resulting from the split of the quadrants. n=7 cells.

- 1.6 Regarding the optimization of the labeling conditions, it appears in Figure 3b that the best DOL was obtained for 16 hours incubation time. How was non-specific labeling affected by incubation time? Based on their data, can the authors present a single set of labeling conditions (concentration, time, etc) which yield the highest DOL with the lowest non-specific background for each probe?

Response 1.6: The Reviewer raised an important point that we decided to address in the discussion section of the revised manuscript as well as by including additional data. Depending on the imaging technique and the target of interest, a higher DOL with increased unspecific background can be favourable over a higher signal-to-background ratio. Therefore, optimal conditions strongly depend on the actual experiments as explained in the manuscript (lines 209-212 and 265-268). Thus, we decided against recommending a single set of labelling conditions. As we observed that unspecific labelling can vary across cell lines, we performed additional experiments to determine the unspecific labelling of Jurkat T cells with Halo-SiR under identical conditions as were used for generating the specific labelling matrix shown in Fig. 3b. This plot is now shown as Supplementary Fig. 10 in the revised manuscript. Finally, we revised the discussion section where we now explain that screening for unspecific labelling in addition to specific labelling is highly recommended (lines 265-276). When taking both parameters into account, intermediate substrate concentrations and incubation times typically lead to optimal target staining for a given imaging technique. We believe that these changes now provide a more comprehensive picture of how ProDOL enables choosing suited labelling conditions.

- 1.7 In Supplementary Figure 4 panels a and b, there is a pronounced dip in the DOL vs incubation time plot at a time of 1 hour. Does this make sense?

Response 1.7: We appreciate the Reviewer noticing that the data presented in Supplementary Fig. 4 of our initial manuscript (Supplementary Fig. 9 in the revised manuscript), shows unexpected staining behaviour for both tags indicating a systematic variation of staining conditions. These experiments were indeed performed in parallel and in the same sample for both tags, however samples for different incubation times were prepared on different days. Thus, the observed dip in all samples with incubation times $t=1$ h were likely caused by deviations during sample preparation for these conditions.

Consistent with this explanation, experiments where we avoid such potential sources of variation by preparing all samples at the same time (shown e.g. in Fig. 3b and Supplementary Fig. 10) do not exhibit comparable phenomena. After thorough discussions, we decided to retain the data in the revised manuscript to raise awareness that such experimental variances, which might easily be overlooked when staining target samples, can easily be detected by performing simultaneous calibration measurements with the ProDOL probe. In the revised manuscript, we now stress the importance of measuring the DOL, with identical staining solutions and under identical conditions to identify such potential variances (lines 268-276). Furthermore, we have revised the caption of the former Supplementary Fig. 4, now Supplementary Fig. 9, clarifying that measurements for SNAP-tag and HaloTag labelling were carried out in the same sample. In addition, we have added a cautionary note to the caption of Supplementary Fig. 9 underlining that ProDOL can be used to identify such deviations and improve the robustness of control experiments. Should Supplementary Fig.9 be deemed overly confusing it can be omitted according to editorial decision.

1.8 *How accurate is the threshold determination / density correction procedure presented in the methods section? Can the authors validate this procedure with experimental data (e.g. with cells expressing very low densities of ProDOL)? Do errors introduced at this step contribute significantly to the variance of the DOL result?*

Response 1.8: The reviewer raised an important point which we addressed by expanding the methods section of the manuscript to explain more clearly how threshold values are computed (lines 660-669). We additionally included a new Supplementary Fig. 2 to illustrate our approach to compute a distance threshold for computing the fraction of colocalizing signals. We agree with the reviewer that the density correction procedure will have an influence on the variance of determined DOLs across cells with large differences in ProDOL probe expression levels. To show that the required density correction of DOL values does not result in biased DOL measurements, we analysed simulated data with variable probe densities representative for the chosen experimental conditions (see Supplementary Fig. 5 for typical probe densities) and variable background levels and found that for all tested scenarios simulated DOL values could be recovered (Supplementary Fig. 4).

1.9 *The new approach to measuring DOL is based on the assumption that the DOL of a protein of interest in its cellular context will be the same as the DOL of the ProDOL probe at the plasma membrane. The authors correctly questioned this assumption in the discussion section of the manuscript, and note that their benchmarking results for Nup107 would support this argument. Could the authors also test this by targeting the ProDOL probe to another region of the cell? Can the authors think of another benchmarking target (e.g. in the nuclear interior, at a mitochondrion, etc.) which could be used to verify the assumption, similar to their Nup107 data? This comment is not necessarily asking for any new data or experiments, but should be seen rather as a question for the authors to comment on, and possibly a suggestion to slightly expand this area of the discussion.*

Response 1.9: We agree with the Reviewer's comment that DOL analysis with ProDOL assumes that comparable DOLs can be achieved for intracellular targets and the ProDOL probe residing on the inner leaflet of the plasma membrane. As described in our response to the request for additional experiments further below (response 2.2), we now provide further support for this assumption by direct comparison of DOLs achieved for ProDOL probe and nuclear pore complex (NPC) labelling via staining in live cells (Fig. 2g). In addition, we added a brief discussion of this issue to the manuscript (lines 279-285). As reviewer #2 provided similar feedback, we kindly refer the reviewer to response 2.2 below for additional remarks on this aspect. There we also discuss our attempts to validate the ProDOL method with an additional intracellular calibration target.

1.10 *The application of the ProDOL method to TCR signaling, presented in the last section of the main text, is extensive and detailed. While I appreciate that this data fully demonstrates the validity of the new approach, adding too much detail here risks losing the reader and detracting from the main message of the paper, which for me seems to me the new method for measuring labelling efficiency. Also, the importance of the new biology which is presented here risks becoming lost because there is simply not sufficient space to do it justice. Is there a way to simplify this section of the manuscript, and Figure 4, while maintaining the demonstration of the ProDOL method in a real biology application?*

Response 1.10: We agree that the T cell experiments presented in the initial version of our manuscript can potentially distract from the main purpose of this manuscript. We have therefore limited the main manuscript to the demonstration of the quantification of SPL-76-Halo during T cell activation in absence and presence of the viral factor Nef and shifted the experiments assessing the phosphorylation state of SLP-76 to Supplementary Note 1 and Supplementary Figs. 11 and 12.

2 Reviewer #2:

2.1 *This manuscript by Tashev et al. presented a new way to evaluate the degree of labeling (DOL) for protein-tag (e.g. SNAP-tag, Halo-tag) labeling. The authors first fused SNAP-tag and Halo-Tag to a membrane anchored reference fluorescent protein (eGFP in this work). The DOL of the protein-tag labeling can then be determined by the percentage of reference fluorescent protein that are colocalized with the labelled protein-tags. Compared to other similar works using antibody or another protein-tag as references, current approach used fused fluorescent protein as reference. It could avoid the effect of unspecific labeling of the reference signal which could result in an underestimation of DOL. The authors showed different DOL of the SNAP-tag and Halo-tag for different labeling approaches (e.g. different dye concentrations, incubation time fixation conditions). Combined with CoPS, the authors also used this approach to measure the protein copy number in T cells. The precise determination of the DOL is very important for quantitative fluorescence microscopy. Although the current approach offers a potentially easy and versatile tool for this purpose, I am not convinced that it is generally applicable to different biological systems without the need for further validation.*

Response 2.1: We thank the Reviewer for critically assessing our manuscript and highlighting its importance for the field of quantitative microscopy. We understand that the general applicability of the method requires further validation as raised also by Reviewer #1. As discussed in depth in response to their comments (1.9) as well as in the following comment below (2.2), we have added data from live cell experiments demonstrating that the labelling efficiency at the plasma membrane and the nuclear pore are comparable and hope that the Reviewer agrees that these strongly support the assumption that the labelling efficiency should be comparable for target proteins throughout the cytoplasm.

2.2 *The DOL calibration probe is evaluated independently with the protein of interest (POI). Therefore, it has to assume that similar DOL of the protein-tag within the calibration probe and POI. Although the author showed similar DOL of the protein-tag with the plasma membrane anchored ProDOL probe and nuclear pore complex, it is still not convinced for me that this method could be generally applicable to other systems. It is well known that many local environmental parameters could affect the maturation of the protein-tags and their accessibility to the fluorophores, such as intra- or extracellular location, fused proteins, and fixation conditions. Especially, the authors also observed significant variations of the DOL across different cell lines as show in Supplementary Fig. 4.*

Response 2.2: As raised by both reviewers (see also our response 1.9), the validation of the assumption that the labelling efficiency for target proteins in different locations within a cell is the same or at least comparable is crucial for the general applicability of the presented approach. Prompted by the reviewers' comments, we performed experiments for further validation using cytoplasmic protein complexes with known stoichiometry.

For such experiments, the difficulty is to identify target proteins which reliably form stable assemblies with defined stoichiometry. We have, for example, undertaken experiments with bacterial GlnA and FtnA protein oligomers which were suggested by others before as suitable calibration targets for quantitative microscopy (Finan et al., *Angew. Chem. Intl. Ed.* 2015). Specifically, we transiently expressed GlnA-HaloTag in HeLa cells and determined the number of fluorophores per GlnA cluster using CoPS analogous to experiments performed with Nup107-Snap presented in Fig. 2 of our manuscript. In these experiments, we found that GlnA consistently formed higher order oligomers under all tested fixation and sample preparation conditions which prevented an unequivocal determination of the DOL using these targets (see figure R2.2 below).

As these findings are in agreement with recently published data by others (Virant et al., *J. Cell Biol.* 2023), we currently do not feel that these complexes meet the standards required for cross-validation and we therefore decided to omit GlnA as a second validation target from the paper.

Figure R2.2: CoPS analysis of GlnA-HaloTag complexes. GlnA-HaloTag was transiently expressed in HeLa cells and sample were processed following a protocol modified from Finan et al. (2015), *Angew. Chem.*). Briefly, samples were labelled with Halo-SiR (10 nM, 1 hr incubation time), fixed with 3.7% PFA and permeabilized with 0.2% TritonX-100. a) Overview scan of cells expressing GlnA-HaloTag after fixation. Inset: Coincidence photon histograms from recorded from indicated clusters and used for CoPS fitting. b) Emitter number per cluster as determined by CoPS (blue). Binomial distributions assuming 12mers and a DOL of 45% (red) or a mixture of 12mers and 2x12mer aggregates assuming a DOL of 34% (orange). Labelling with Halo-SiR. 346 clusters from n=22 cells.

In a second line of experiments, we instead opted to re-use Nup107 as calibration target which we previously used for cross-validating ProDOL (Fig. 2). In these cross-validation experiments, we performed staining with SNAP-AF647 in fixed and permeabilised cells which is required due to the lack of cell permeability of SNAP-AF647, a dye essential for cross-validation by ELE.

To address the reviewer's concern that the subcellular environment of fixation might alter the maturation or activity of protein tags, we instead performed labelling with SNAP-SiR in live U2-OS cells expressing the ProDOL probe or Nup107-SNAP (lines 172-175). We then proceeded by fixing cells and imaging them according to the ProDOL approach or (for cells expressing the ProDOL probe) or by CoPS (for cells expressing Nup107-SNAP). These experiments revealed no statistically significant difference in the labelling efficiencies for SNAP-tag residing at the plasma membrane (ProDOL probe) and nucleoporins embedded in the protein dense NPC structure deep within the cytoplasm.

We revised the manuscript by presenting this data in Fig. 2g and by more clearly stating that the ProDOL approach in its current state is validated only for assessing the DOL of proteins residing within or exposed to the cytoplasm of mammalian cells (lines 279-282). We also added a comment that further validations will be needed to establish the ProDOL approach for other subcellular structures once suitable validation targets of other subcellular structures become available (lines 282 – 285).

2.3 *In the main text line 241-244, the number of the phospho-SLP-76 was determined using CoPS. The author simply corrected the number with dye to antibody ratio. However, the binding efficiency of the antibody to phospho-SLP-76 was not known. It is therefore difficult to determine the copy number of phospho-SLP-76.*

Response 2.3: We agree with the reviewer that for immunolabeling, the binding efficiency of the used antibody will have to be taken into account. However, there are a multitude of issues arising from the determination of antibody binding efficiency, something which we have now added to our discussion in lines 297-306 in the revised manuscript. In fact, this was one of the main reasons for us to confine ProDOL calibration to protein tags. Yet, even with an unknown binding efficiency of the antibody to phospho-SLP-76, it is still possible to quantify relative changes in the degree of phosphorylation across different biological conditions. As the analysis of SLP-76 phosphorylation is limited by the lack of knowledge about the antibody's binding efficiency, we now refer to these experiments as 'semi-quantitative' in the main text (lines 244-254).

As detailed in response 1.10 to Reviewer #1, we have also decided to shift the analysis of SLP-76 phosphorylation to a Supplementary Note 1 to sharpen the focus of the main text.

2.4 *The effective labeling efficiency used in Ref. 10 is defined by the fraction of target proteins that carry a fluorophore that is detected as a usable localization. This parameter takes into account bleaching of the fluorophores or dim and nonfunctioning fluorophores during SMLM imaging. In theory, this value should be slightly lower than the DOL defined in this work.*

Response 2.4: In STORM microscopy, the initial bleaching is used to drive most fluorophores into a dark state. While one could assume a higher proportion of photobleaching as compared to regular TIRF illumination as carried out in the ProDOL experiments. However, STORM imaging critically relies on high label densities which, at least to some extent, speaks against massive photobleaching. Moreover, we expect photobleaching to also happen in ProDOL experiments. Unfortunately, the mechanisms of photobleaching are still not fully understood and depend on various factors including the excitation power, the fluorescent label, the local environment of the fluorophore (e.g. aromatic amino acids), as well as the use of photostabilising and/or switching buffers. Together, these effects prevent a clearcut prediction of the extent of photobleaching. Thus, a robust comparison of photobleaching in the ELE with other methods would require a lot of careful experiments accounting for all the parameters listed above which are beyond the presented work. Therefore, we stick to our experimental

observation that the labelling efficiencies the discussed methods seem to be comparable with no significant difference.

- 2.5 *The cut-off T to determine the colocalization is a bit too complex to me. Is it necessary to do so? Since the expression level of the ProDOL probe can be adjusted so that the distance between individual ProDOL probe is larger than the diffraction limit, it is actually quite easy to set the cut-off T less than the diffraction limit to determine whether reference signals and protein-tag signals are colocalized or not.*

Response 2.5: We agree with Reviewer 2 that in principle it is possible to perform the ProDOL analysis with a fixed distance cut-off T based on the diffraction limit for a given microscope. Indeed, values for T are typically well below the theoretical diffraction limit and were found to be between 105-190 nm for the microscopes used for data acquisition. However, multiple factors including e.g. signal localisation uncertainty, probe density and the alignment quality of multispectral images will impact T. For this reason, we feel that providing a routine to determine T for individual experiments increases the robustness of the ProDOL data processing pipeline. In line with the Reviewer's comment 2.7 below, we rewrote the methods section of our revised manuscript to allow readers to better understand the individual data processing steps (lines 660-669). To illustrate the rationale behind the procedure to determine T, we added an additional Supplementary Fig. 2 which shows how T is computed. In the same figure, we also show a typical colocalization score (Z) vs. spatial tolerance (t) plot used for determining T.

- 2.6 *Since most experiments were performed on the plasma membrane near the coverslip, did the authors investigate the effect of the non-specific binding of the dyes on the coverslip?*

Response 2.6: Unlike unspecific background signal due to autofluorescence (see Supplementary Fig. 5b in revised manuscript), unspecific binding of dyes to the coverslip will indeed contribute to the signal detected in the ProDOL assay. To assess this, we measure unspecific dye binding on the coverslip and within cells by determining the trapped dye density in our samples using the LynG probe which lacks both SNAP-tag and HaloTag (see e.g. Fig. 3a, Supplementary Figs. 8, 9, 10). Typically, unspecific binding to the coverslip increases with increasing incubation time and substrate concentration as shown in Supplementary Fig 9d. In our data analysis pipeline, we include two critical steps to minimize the influence of non-specific signals onto computed DOLs. First, we segment the area covered by cells in input images and only take signals detected within segmented cells into account. Second, by computing a stringent distance cut-off value T for computing the fraction of co-localizing reference and tag signals (see our previous to reviewer question R2.5), we require close proximity between reference signals and protein tag signals. At a low density of unspecific dye deposition, this means that the vast majority of unspecifically deposited dyes will not have an impact on the computed DOL using the ProDOL probe. To illustrate this, we now include an analysis that computes the apparent DOL observed in Jurkat T cells expressing the LynG control probe and stained with Halo-SiR under varying staining conditions (Supplementary Fig. 10b). This analysis allows us to show that the impact of unspecific dye deposition on the coverslip or within cells has a negligible effect on measured DOLs.

- 2.7 *Many parameters of the three equations in the methods were not defined. Equation 2 should be re-written.*

Response 2.7: We have revisited the manuscript and Supplemental Information and incorporated all missing information related to the underlying mathematical aspects. We also expanded and rephrased the methods description to facilitate ease of reproduction by users

(lines 660-679). We trust that these additions address the concerns raised. If there are any specific areas the Reviewer would like us to revisit or elaborate further, please do not hesitate to let us know. We remain committed to ensuring the clarity and reproducibility of our work.

2.8 *Line 216-218, the authors mentioned that only cells that adhered to the cover slide within the first 3 seconds were observed, and all other cells washed away. I am curious how this can be performed. Can the authors give more information about this?*

Response 2.8: For these experiments, we have used a modified protocol previously described in Abraham et al. J. Immunol. 2012 (Ref. 27 in the revised manuscript). Here, cells were deposited in high concentration (2×10^6 / 100 μ l) onto the cover slip, before being taken out three seconds later. The cells that have already attached to the coated coverslip are then incubated in PBS, before being fixed in 4% PFA after the indicated activation time. In combination with the dense coverslip coverage with the activating anti-CD3 antibody, this approach allows us to achieve a defined temporal window of activation. More information about the protocol is given in the methods section (now in lines 480-484 of our revised manuscript).

Decision Letter, first revision:

Dear Dirk,

Thank you for submitting your revised manuscript "ProDOL: A general method to determine the degree of labelling for staining optimisation and molecular counting." (NMETH-A53770A). It has now been seen by the original referees and their comments are below. The reviewers find that the paper has improved in revision, and therefore we'll be happy in principle to publish it in Nature Methods, pending minor revisions to comply with our editorial and formatting guidelines.

TRANSPARENT PEER REVIEW

Nature Methods offers a transparent peer review option for new original research manuscripts submitted from 17th February 2021. We encourage increased transparency in peer review by publishing the reviewer comments, author rebuttal letters and editorial decision letters if the authors agree. Such peer review material is made available as a supplementary peer review file. Please state in the cover letter 'I wish to participate in transparent peer review' if you want to opt in, or 'I do not wish to participate in transparent peer review' if you don't. Failure to state your preference will result in delays in accepting your manuscript for publication.

Please note: we allow redactions to authors' rebuttal and reviewer comments in the interest of confidentiality. If you are concerned about the release of confidential data, please let us know specifically what information you would like to have removed. Please note that we cannot incorporate redactions for any other reasons. Reviewer names will be published in the peer review files if the reviewer signed the comments to authors, or if reviewers explicitly agree to release their name. For more information, please refer to our FAQ page.

ORCID

Sincerely,
Rita

Rita Strack, Ph.D.
Senior Editor
Nature Methods

Reviewer #1 (Remarks to the Author):

In response to the reviewers' comments, Tashev et al. have fully addressed the questions which were raised in the first review. I am satisfied that this manuscript demonstrates a novel and powerful technique for in situ measurement of the DOL for various labeling methods used in fluorescence microscopy, and that it makes an important contribution to the field. The manuscript is well written and thoroughly addresses the practical, technical, and quantitative details of the new method and its results. I recommend publication of the revised manuscript.

Reviewer #1 (Remarks on code availability):

I have reviewed the Github repository and the documentation. Both the code and the documentation appear to be well organized and complete.

Reviewer #2 (Remarks to the Author):

The authors performed additional experiments to address my concerns of the effects of real cellular environments could affect the degree of labeling (DOL), which I really appreciate and believe is very crucial for the current work. The authors also add some warnings/discussion about applying the current method to different proteins of interest (line 279-285) at different locations. This is also necessary since the current method can not be used without a warranty. Although I think it is still quite challenging to quantify the DOL of protein tag in-situ without the disturbance to the proteins of interest, this work provides a simple, and clean method to semi quantify the DOL of protein tag. DOL of protein tags is important for quantitative fluorescence imaging. I believe current work would attract broad interest in the field and recommend it for publication.

Author Rebuttal, first revision:

Response to referees

Reviewer #1: (Remarks to the Author):

In response to the reviewers' comments, Tashev et al. have fully addressed the questions which were raised in the first review. I am satisfied that this manuscript demonstrates a novel and powerful technique for in situ measurement of the DOL for various labeling methods used in fluorescence microscopy, and that it makes an important contribution to the field. The manuscript is well written and thoroughly addresses the practical, technical, and quantitative details of the new method and its results. I recommend publication of the revised manuscript.

Reviewer #1 (Remarks on code availability):

I have reviewed the Github repository and the documentation. Both the code and the documentation appear to be well organized and complete.

Reviewer #2 (Remarks to the Author):

The authors performed additional experiments to address my concerns of the effects of real cellular environments could affect the degree of labeling (DOL), which I really appreciate and believe is very crucial for the current work. The authors also add some warnings/discussion about applying the current method to different proteins of interest (line 279-285) at different locations. This is also necessary since the current method can not be used without a warranty. Although I think it is still quite challenging to quantify the DOL of protein tag in-situ without the disturbance to the proteins of interest, this work provides a simple, and clean method to semi quantify the DOL of protein tag. DOL of protein tags is important for quantitative fluorescence imaging. I believe current work would attract broad interest in the field and recommend it for publication.

Response:

We thank both reviewers for their approval of the changes made in the revised manuscript and their helpful comments on the initial manuscript.

Final Decision Letter:

Dear Dirk,

I am pleased to inform you that your Article, "ProDOL: A general method to determine the degree of labelling for staining optimisation and molecular counting.", has now been accepted for publication in Nature Methods. The received and accepted dates will be Sept 8, 2023 and June 24, 2024. This note is intended to let you know what to expect from us over the next month or so, and to let you know where to address any further questions.

Over the next few weeks, your paper will be copyedited to ensure that it conforms to Nature Methods style. Once your paper is typeset, you will receive an email with a link to choose the appropriate publishing options for your paper and our Author Services team will be in touch regarding any additional information that may be required. It is extremely important that you let us know now whether you will be difficult to contact over the next month. If this is the case, we ask that you send us the contact information (email, phone and fax) of someone who will be able to check the proofs and deal with any last-minute problems.

Please note that *Nature Methods* is a Transformative Journal (TJ). Authors may publish their research with us through the traditional subscription access route or make their paper immediately open access through payment of an article-processing charge (APC). Authors will not be required to make a final decision about access to their article until it has been accepted. Find out more about Transformative Journals

You may wish to make your media relations office aware of your accepted publication, in case they consider it appropriate to organize some internal or external publicity. Once your paper has been scheduled you will receive an email confirming the publication details. This is normally 3-4 working days in advance of publication. If you need additional notice of the date and time of publication,

please let the production team know when you receive the proof of your article to ensure there is sufficient time to coordinate. Further information on our embargo policies can be found here: <https://www.nature.com/authors/policies/embargo.html>

If you are active on Twitter/X, please e-mail me your and your coauthors' handles so that we may tag you when the paper is published.

Best regards,
Rita

Rita Strack, Ph.D.
Senior Editor
Nature Methods